# Blood pressure variability compromises vascular function in middle-aged mice

Perenkita J Mendiola, Philip O'Herron, Kun Xie, Michael W Brands, Weston Bush, Rachel E Patterson, Valeria Di Stefano, Jessica A Filosa*

Department of Physiology, Medical College of Georgia, Augusta University, Augusta, United States

## eLife Assessment

This is an **important** study that demonstrates that blood pressure variability impairs myogenic tone and diminishes baroreceptor reflex. The study also provides evidence that blood pressure variability blunts functional hyperemia and contributes to cognitive decline. The evidence is **compelling** whereby the authors use appropriate and validated methodology in line with or more rigorous than the current state-of-the-art.

*For correspondence:
JFILOSA@augusta.edu

Competing interest: The authors declare that no competing interests exist.

**Abstract** Blood pressure variability (BPV) has emerged as a significant risk factor for cognitive decline and dementia, independent of alterations in average blood pressure (BP). However, the impact of large BP fluctuations on neurovascular function remains poorly understood. In this study, we developed a novel murine model of BPV in middle-aged mice using intermittent angiotensin II infusions. Radio telemetry confirmed that 24 hr BP averages in BPV mice remained comparable to controls, demonstrating BPV in the absence of hypertension. Chronic (20–25 days) BPV resulted in a blunted bradycardic response and cognitive deficits. Two-photon imaging revealed heightened pressure-evoked constrictions (myogenic response) in parenchymal arterioles of BPV mice. While sensory stimulus-evoked dilations (neurovascular coupling) were amplified at higher BP levels in control mice, this pressure-dependent effect was abolished in BPV mice. Our findings indicate that chronic BP fluctuations impair vascular function within the neurovascular complex and contribute to cognitive decline, emphasizing BPV as a critical factor in brain health.

## Introduction

Dementia, the seventh leading global cause of mortality, presents as the predominant neurodegenerative complication in the elderly population. Notably, hypertension emerges as a primary risk factor for dementia (*Baggeroer et al., 2024*; *Iadecola et al., 2016*). Blood pressure (BP) regulation involves a complex interplay of central and peripheral mechanisms, finely attuned to acute and chronic stressors in maintaining physiological ranges. Perturbations in vascular function and structure (e.g. reactivity, vascular stiffness, *Boutouyrie et al., 2021*), alterations in baroreflex sensitivity (*Hesse et al., 2007*), and enhanced sympathetic activation (*de Leeuw et al., 2017*) are recognized factors that disrupt BP balance. Blood pressure variability (BPV) (an index of BP *fluctuation or variation*), particularly in midlife, increases the risk for cognitive decline (*Nagai et al., 2012*), end-organ damage (*Vishram et al., 2015*), and cardiovascular events (*Vishram et al., 2015*; *Ernst et al., 2020*). Increased BPV also serves as an early predictor of hypertension (*Böhm et al., 2015*). BPV delineates fluctuations across various measurement periods from very short-term (beat-to-beat) to long-term (months or years, *Parati et al., 2018*).

Given evidence that elevated BPV can precede hypertension (*Özkan et al., 2022*), a modifiable risk factor in the setting of vascular cognitive impairment and dementia, understanding the drivers and cellular targets of BPV is essential. Unlike hypertension, BP fluctuations often elude detection in screening practices conducted in clinical settings, which rely on singular or averaged BP measurements (*Schutte et al., 2022*). The absence of a gold standard for BPV indexing (*Del Giorno et al., 2019*) and limited insights into the magnitude and frequencies of impactful BP fluctuations in disease conditions further compound this issue. Additionally, the absence of suitable animal models has limited the understanding of how this overlooked variable may impact brain health. In this study, we sought to establish an innovative murine model of BPV and to interrogate the impact of large BP fluctuations on cardiovascular and neurovascular outcomes.

Cerebral blood flow (CBF) is tightly regulated by diverse signaling processes, including the interaction of cellular elements comprising the neurovascular complex, which include endothelial cells, vascular smooth muscle cells, pericytes, astrocytes, neurons, and microglia (*Iadecola, 2017*; *Schaeffer and Iadecola, 2021*). Mechanisms governing CBF operate under both steady-state conditions (e.g. cerebral autoregulation (CA), humoral processes, chemoregulation) and in response to local neuronal activity (e.g. neurovascular coupling (NVC); *Claassen et al., 2021*). Evidence suggests that sustained hypertension adversely affects the functional integrity of the neurovascular complex (*Capone et al., 2012*; *Faraco et al., 2016*; *Santisteban et al., 2023*), compromises cerebral perfusion (*Capone et al., 2012*; *Faraco et al., 2016*), and contributes to cognitive decline (*Faraco et al., 2016*). However, the impact of large BP fluctuations on steady-state or activity-evoked CBF changes remains poorly understood.

Chronic hypertension leads to adaptive processes (e.g. vascular remodeling, *Izzard et al., 2006*; *Pires et al., 2015*) and a rightward shift in the cerebral autoregulation (CA) curve (*Iadecola and Davisson, 2008*), heightening vulnerability to ischemia at low BP while shielding the brain from hyperperfusion at high pressure. In humans, hypertension and BPV have been associated with white matter hyperintensities (*van Dijk et al., 2004*; *Dufouil et al., 2001*; *Zhang et al., 2022* and microbleeds *Elmståhl et al., 2019*; *Reddy and Savitz, 2020*; *Liang et al., 2022*). Intriguingly, the impact of BPV (prior to hypertension onset) on cerebrovascular function and neurovascular outputs has received little attention.

Using an innovative murine model of BPV, in the absence of overt hypertension, combined with in vivo two-photon imaging, we show that chronic BP fluctuations lead to microvascular dysfunction (e.g. enhanced parenchymal arteriole myogenic responses) and a blunted NVC response. In addition, mice subjected to chronic BPV showed poor cognitive performance. These findings underscore the pivotal role of dysregulated BP events in brain health and function.

## Results

### Pulsatile Ang II infusion-induced changes in blood pressure

We developed a novel murine model of high BPV using the experimental design shown in *Figure 1A*. The protocol consisted of a baseline phase (~5 days), corresponding to saline infusions in all mice, followed by a treatment phase (~20–25 days), where mice were either left on saline (control group) or subjected to robust BP transients induced via Ang II infusions (BPV group). Subcutaneous pumps were programmed to infuse 2 μL 6–8 times per day, with infusion periods corresponding to 1 hr every 3–4 hr (depending on the number of pulses), *Figure 1B*, right panel.

During the pump-*off* periods, BP corresponded to 97±2.0 mmHg and 91±1.9 mmHg for MAP, 113±2.6 mmHg and 105±2.6 mmHg for SBP, 80±1.7 mmHg, and 78±1.7 mmHg for DBP, and 34±1.3 mmHg and 27±1.8 mmHg for PP, for mice with pumps infusing saline vs. Ang II, respectively (*Figure 1C*). During the pump-*on* period, saline infusion did not change BP. However, Ang II infusions evoked significant BP transients corresponding to Δ37±3.9 mmHg (p<0.0001) for MAP, Δ70±3.0 mmHg (p<0.0001) for SBP, Δ29±4.1 mmHg (p<0.0001) for DBP, and Δ28±3.0 mmHg (p<0.0001) for PP, *Figure 1C*. Note that large dynamic activity-induced BP changes were apparent during both saline- and Ang II-infusions, *Figure 1B*. Despite robust BP transients, 20 days of treatment minimally affected the 24 hr averages (vs Baseline) in all cardiovascular variables, *Figure 1D*. The delta BP (Baseline vs Day 20) for MAP was −1±2.0 mmHg and 6±2.4 mmHg for control and BPV mice, respectively (*Figure 1D*). While the SBP in BPV mice significantly increased Δ9±2.9 mmHg (p=0.0088),

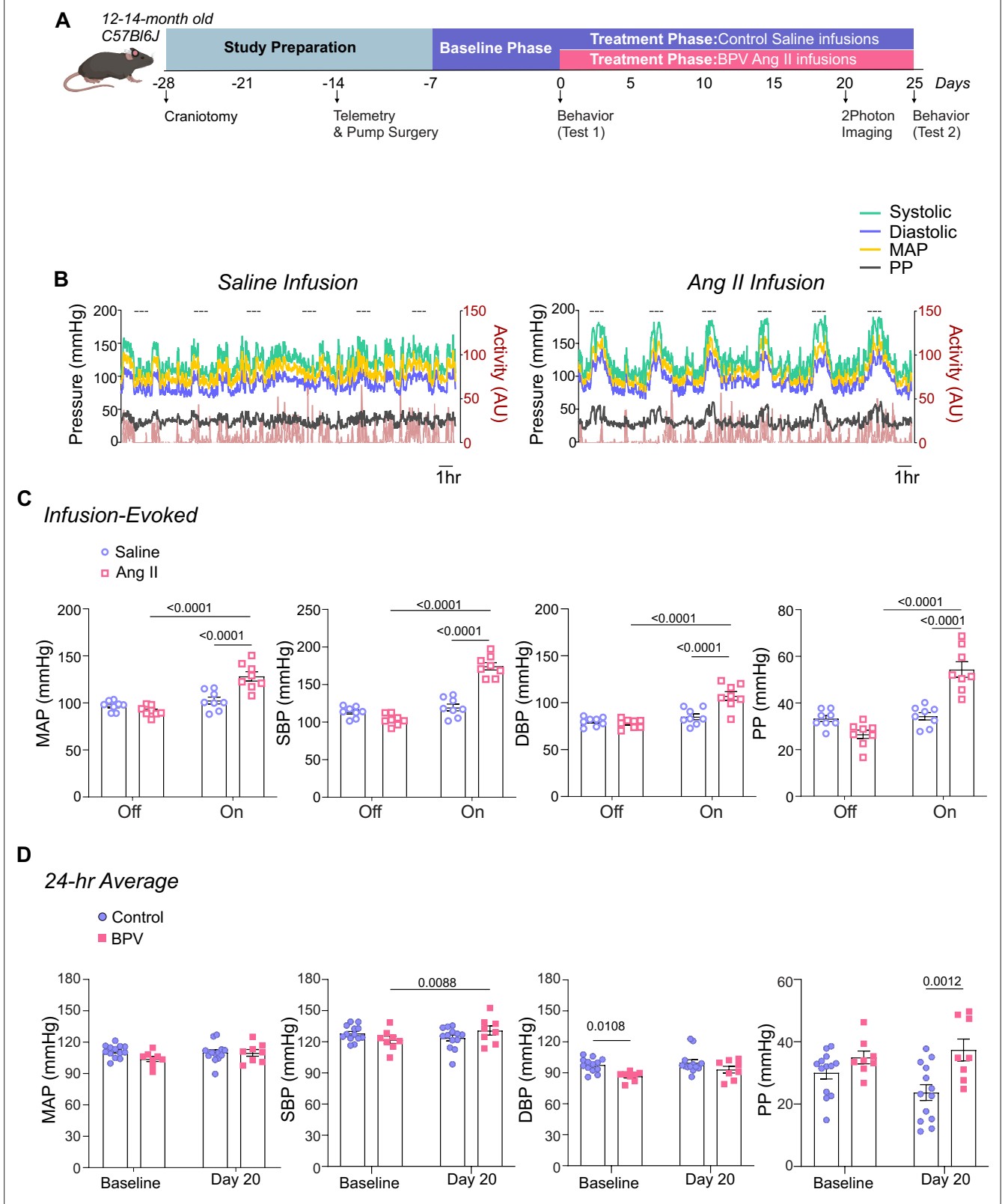

**Figure 1.** Experimental design and Ang II infusion effects on mean arterial pressure. (**A**) Schematic of the experimental design for middle-aged C57BL/6 J male mice, including implantation of a chronic cranial window (–28 days from treatment onset) and an infusion pump with telemetry (–7 days from treatment onset). Infusion pumps were programmed for intermittent delivery of saline (baseline phase) or angiotensin II (Ang II) (treatment phase) at a rate of 18.4 µg/day, administered in 1 hr intervals every 3–4 hr. In-vivo two-photon (2 P) imaging sessions were performed approximately 20 days

*Figure 1 continued on next page*

*Figure 1 continued*

into the treatment phase. Behavioral tests were conducted during the baseline period (Test 1) and 25 days into the treatment phase (Test 2). Created with BioRender.com. (**B**) Representative raw traces depicting minute-to-minute, 24 hr MAP, SBP, DBP, PP, and Activity levels during intermittent saline infusions (left) and Ang II infusions (right). Dashed lines indicate periods of active infusion (1 hr) or when the pump is *on*. (**C**) Average MAP, SBP, DBP, and PP (5 min) measured during infusion *Off* versus *On* conditions. (**D**) 24 hr average MAP, SBP, DBP, and PP during a 5 day saline infusion phase (Baseline) and on Day 20 of Ang II treatment. Two-way ANOVA repeated measures followed by Sidak's multiple comparisons test (**C**, n=8 BPV mice) (**D**, n=13 control mice, n=8 BPV mice). Ang II = Angiotensin II, BPV = blood pressure variability, DBP = diastolic blood pressure MAP = mean arterial pressure, *P*=pulse pressure, SBP = systolic blood pressure.

The online version of this article includes the following figure supplement(s) for figure 1:

**Figure supplement 1.** One-hour averages of MAP over 24 hr-Circadian Profile.

mice remained within the lower range of high BP (131±4.4 mmHg). Additionally, prior to treatment, the baseline DBP in BPV was significantly lower than in control mice (p=0.0108), *Figure 1D*.

Blood pressure exhibits a circadian rhythm that is not discernible in minute-to-minute traces, (*Figure 1B*) but remains intact in both controls and BPV mice when analyzed in 1 hr averages, *Figure 1—figure supplement 1A–D*. Thus, cardiovascular variables were assessed during the active (dark) and inactive (light) phases of the mouse 12:12 hr cycle. Consistent with the unchanged 24 hr average for MAP, there were minimal changes in the averaged BP corresponding to the daily inactive and active periods for control and BPV mice, *Figure 2A–D*. Compared to controls, the DBP of the BPV group tended to be lower but not significant (p=0.09), *Figure 2C*. However, beginning on day 7 of treatment, BPV (vs control) mice exhibited significantly higher PP during the inactive period, *Figure 2D*. These data suggest a compensatory mechanism that maintains MAP within physiological ranges, albeit with elevated PP in BPV mice.

## Pulsatile Ang II infusion induced high BPV

Having achieved minimal changes in the 24 hr average MAP, yet with prominent BP fluctuations, we quantified blood pressure variability (BPV) over time. Two indices of BPV were assessed: the average real variability (ARV), defined as the absolute difference between consecutive BP measurements, and the coefficient of variation (CV), defined as the ratio of the standard deviation to mean BP. A significant increase in BPV was observed in the BPV group over time relative to baseline, despite stable 24 hr BP averages. During the active period, ARV was significantly increased by day 3 and remained increased throughout the protocol, reaching 16.4±0.66 mmHg (p=0.04) for SBP and 8.12±0.71 mmHg (p=0.04) for PP, *Table 1A*. Similarly, during the inactive period, ARV was significantly increased by day 3 and persisted, with values of 13.4±0.33 mmHg (p<0.01) for MAP, 17.3±0.48 mmHg (p=0.02) for SBP, 11.3±0.47 mmHg (p<0.01) for DBP, and 7.9±0.55 mmHg (p=0.04) for PP (*Table 1B*) with SBP as the main variable driving BPV increases in this model. BPV indices using CV, based on hourly BP averages over 24 hr, were significantly increased from day 1 of treatment across all cardiovascular variables during both the inactive and active periods, except for DBP of the active period, which increased on day 3, *Supplementary file 1*.

## Autonomic function in BPV mice

To compare pressure-induced bradycardic responses between BPV and control mice at both early and later treatment stages, a cohort of control mice received Ang II infusion on days 3–5 (early phase) (*Figure 3—figure supplement 1*) and days 21–25 (late phase), thereby transiently increasing BP. Ang II evoked significant increases in SBP in both the control and BPV groups which were accompanied by a pronounced bradycardic response, *Figure 3A*. For a single infusion-evoked BP peak, changes in HR were -Δ291±34 bpm (p<0.0001) and -Δ304±17 bpm (p<0.0001) for control and BPV mice, respectively (*Figure 3B*). While each Ang II infusion evoked significant bradycardic responses, the 24 hr HR average was not significantly altered, corresponding to 612±7 and 585±12 bpm for baseline and 570±25 and 546±8 bpm at day 20, for control and BPV mice, respectively, *Figure 3C*.

To assess whether chronic BP fluctuations affect the baroreflex response, we compared the slopes extracted from linear regression of SBP vs HR, focusing on within-group differences, *Figure 3D–G*. Comparisons were conducted over a 100 min window surrounding a single Ang II-evoked BP pulse (20 min before and after 60 min pulse), occurring during both the active and inactive periods, as well as between the early and late treatment phases. Linear regression analysis of SBP vs HR during Ang

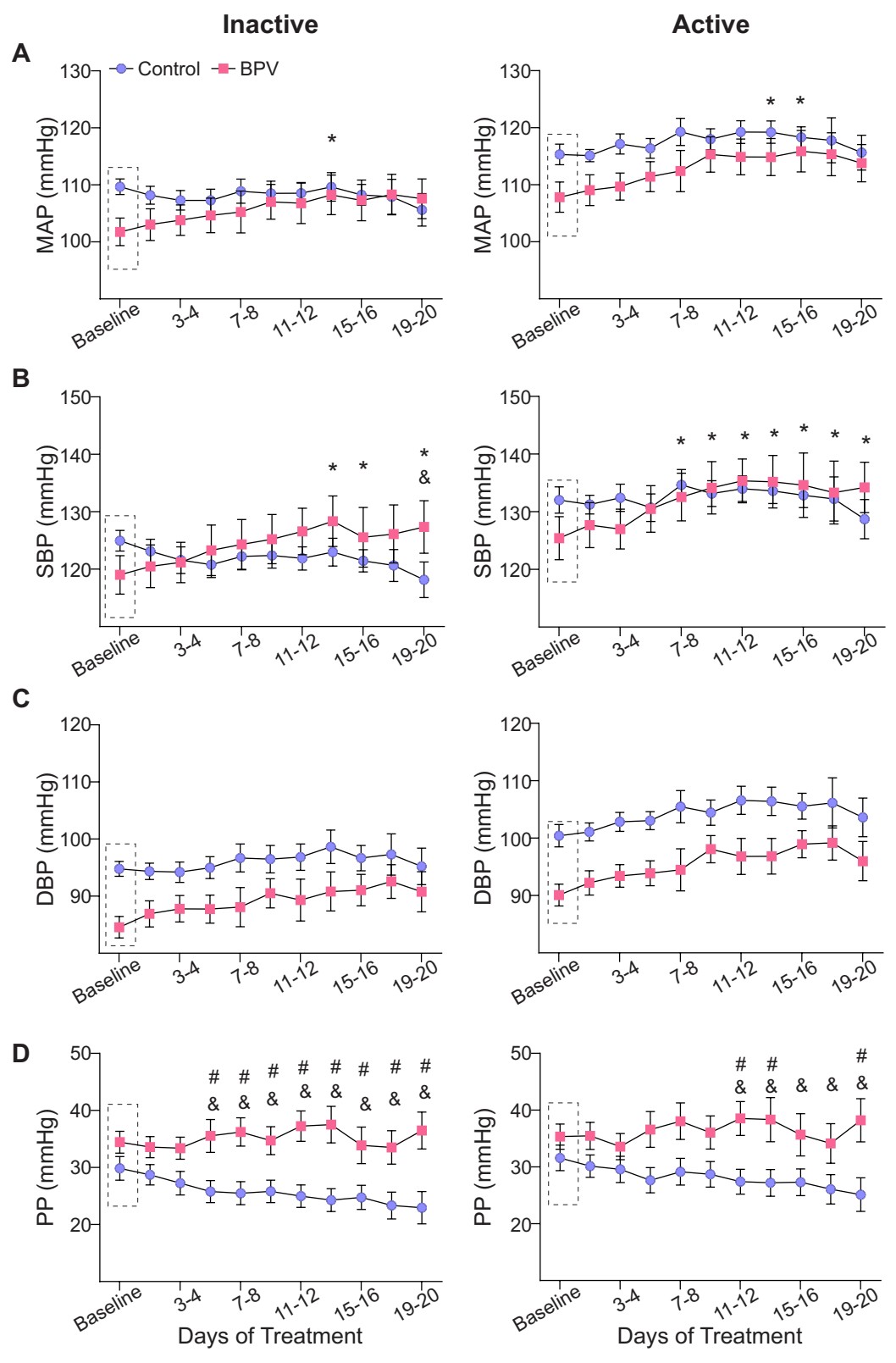

**Figure 2.** Effects of pulsatile BP on average 12:12 hr cardiovascular variables. (**A**) Summary data of two-day averages of MAP, SBP (**B**), DBP (**C**), and PP (**D**) during the inactive (daylight) period (left column) and the active (nighttime) period (right column). Dashed rectangles indicate the baseline period (~5-day saline infusion). '*' and '&' denote within group comparisons ($p < 0.05$ vs Baseline) for BPV and control, respectively. '#' denotes $p < 0.05$

*Figure 2 continued on next page*

*Figure 2 continued*
between groups comparisons. Two-way ANOVA repeated measures followed by Sidak's comparison test; n = 13 control mice, n = 8 BPV mice. BPV=blood pressure variability, DBP=diastolic blood pressure, MAP=mean arterial pressure, PP=pulse pressure, SBP=systolic blood pressure.

II infusion revealed no significant differences in the control group over time (Early vs Late), *Figure 3D and F*. However, the BPV group exhibited a significant reduction in the bradycardic response, indicated by flatter negative slopes during both the inactive (p<0.0001) and active periods (p<0.001) (*Figure 3E and G*). These data suggest that chronic (~25 days) BPV diminishes bradycardic responses, indicative of suppressed autonomic function.

## Pulsatile norepinephrine infusion induced BPV

To determine whether the observed cardiovascular effects of BPV were limited to Ang II, norepinephrine (NE, 45 µg/kg/min) was utilized as an alternative agent to induce BP variations, *Figure 4*. The NE dosage was titrated to match Ang II-induced BP transient pulse levels (150–200 mmHg, SBP) using the same subcutaneous delivery protocol (pump on for 1 hr every 4 hr) resulting in 6 pulses/day, *Figure 4A*. Similar to Ang II-treated mice, NE-treated mice exhibited minimal changes in the 24 hr average MAP throughout the 20 day treatment, *Figure 4B*. While there was a trend towards increased ARV (p=0.08), it did not reach significance, *Figure 4C* left panel. However, CV, another BPV index, was significantly increased by day 3 (p=0.02) of the treatment, *Figure 4C* right panel.

In contrast to the bradycardic response observed during Ang II-induced BP elevations, NE-induced BP pulses triggered a tachycardic response (*Figure 4A*), with significant positive correlation between SBP and HR appearing during the early phase in the treatment (Inactive period, p<0.0001; Active period, p=0.0002), *Figure 4D*. These data validate the intermittent infusion protocol as an effective method for inducing significant BPV while preserving normal 24 hr MAP. However, due to fundamental differences in bradycardic responses and the presence of off-target effects, the physiological actions of this vasopressor agent are not directly comparable to those of Ang II. These results underscore the necessity of careful vasopressor selection to mitigate off-target effects.

## Chronic BPV enhanced parenchymal arteriole myogenic responses

CBF regulation involves complex processes that maintain the metabolic needs of working neurons. These include mechanisms active at baseline (e.g. cerebral autoregulation *Claassen et al., 2021* and endothelial cell-mediated signaling *Longden et al., 2017*) and those involved in activity-mediated increases in CBF or neurovascular coupling (NVC) (*Iadecola, 2017*). To assess if BPV impacts the functional integrity of the neurovascular complex, we measured parenchymal arteriole responses to acute BP increases and determined if BP itself altered the sensory-evoked arteriole response. For these studies, in vivo two-photon imaging was combined with BP measurements via telemetry, *Figures 5 and 6*.

To determine the impact of acute BP changes on the microcirculation, we simultaneously tracked parenchymal arteriole diameter changes before pump infusion onset (Low BP), and during Ang II infusions (High BP), *Figure 5E*. For these experiments, and prior to the imaging session, the pump reservoir for the control group was switched from saline to Ang II. Imaging was conducted at similar cortical depths (~layer 1/2 border of the somatosensory cortex) corresponding to 106±11.6 µm and 102±4.3 µm for control and BPV mice, respectively, *Figure 5B*. Arteriole diameters measured at baseline (Low BP) were comparable between groups, corresponding to 18.6±1.1 µm and 19±1.0 µm for controls and BPV mice, respectively, *Figure 5C*. Upon mild sedation resting BP dropped by –23±3.6 mmHg and –31±4.4 mmHg for the control and BPV group, respectively (not shown). However, BP values remained within the putative cerebral autoregulation plateau range throughout two-photon imaging sessions and were comparable between groups. During the pump infusion period (High BP), the BP increased from 68±4.2–102±5.9 mmHg for controls (p<0.0001) and from 59±4.5–96±6.4 mmHg for BPV mice (p<0.0001), *Figure 5D*.

To quantify in vivo pressure-evoked diameter changes (myogenic responses), we compared the slopes extracted from the linear regression of MAP vs parenchymal arteriole diameter measured throughout the acquisition session, *Figure 5E–G*. The steeper negative slope observed in BPV mice compared to controls indicates significantly increased parenchymal arteriole reactivity to BP

**Table 1.** Ang II-induced increases in blood pressure variability (BPV).

(A-B) Summary data of calculated two-day average real variability of MAP, SBP, DBP, and PP during the active period (A) and the inactive period (B). Baseline corresponds to (~5-day saline infusion) and '*' denotes within group comparisons (p<0.05 vs Baseline) and '#' denotes p<0.05 between groups comparisons. Two-way ANOVA repeated measures followed by Dunnett's comparison test; n = 13 control mice, n = 8 BPV mice.

**A. ARV mmHg ±SEM (Active)**

| Days | MAP Control | MAP BPV | SBP Control | SBP BPV | DBP Control | DBP BPV | PP Control | PP BPV |
|---|---|---|---|---|---|---|---|---|
| Baseline | 12.44 ± 1.37 | 10.94 ± 0.55 | 14.15 ± 1.49 | 13.13 ± 0.66 | 11.53 ± 1.44 | 9.04 ± 0.54 | 4.67 ± 0.4 | 5.47 ± 0.46 |
| 1–2 | 12.91 ± 1.29 | 10.99 ± 0.35 | 14.86 ± 1.28 | 14.56 ± 0.44 | 11.7 ± 1.48 | 9.29 ± 0.41 | 5.12 ± 0.4 | 7.67 ± 0.69 |
| 3–4 | 12.44 ± 0.66 | 12.51 ± 0.43 | 14.4 ± 0.67 | 16.4 ± 0.66 * | 11.12 ± 0.83 | 10.35 ± 0.45 * | 5.03 ± 0.33 | 8.12 ± 0.71 *# |
| 5–6 | 11.78 ± 0.52 | 12.8 ± 0.39 | 13.64 ± 0.61 | 16.44 ± 0.53 *# | 10.71 ± 0.73 | 10.54 ± 0.45 *# | 4.89 ± 0.32 | 7.99 ± 0.69 *# |
| 7–8 | 12.35 ± 0.61 | 12.85 ± 0.41 * | 14.02 ± 0.67 | 16.73 ± 0.43 *# | 11.37 ± 0.81 | 10.61 ± 0.55 *# | 5.22 ± 0.42 * | 8.34 ± 0.74 *# |
| 9–10 | 12.34 ± 0.69 | 12.84 ± 0.4 | 13.88 ± 0.75 | 16.37 ± 0.72 | 11.32 ± 0.79 | 10.82 ± 0.24 | 4.95 ± 0.39 | 7.86 ± 0.65 *# |
| 11–12 | 13 ± 0.76 | 12.94 ± 0.76 * | 14.59 ± 0.88 | 16.87 ± 0.7 * | 11.93 ± 0.93 | 10.59 ± 0.81 * | 5.05 ± 0.34 * | 8.25 ± 0.78 # |
| 13–14 | 12.26 ± 0.84 | 12.9 ± 0.42 * | 13.94 ± 0.84 | 16.8 ± 0.52 * | 11.17 ± 0.95 | 10.4 ± 0.68 * | 5.07 ± 0.39 * | 8.45 ± 0.83 # |
| 15–16 | 13.27 ± 0.59 | 13.91 ± 1.02 | 15.11 ± 0.81 | 17.5 ± 0.87 * | 11.72 ± 0.82 | 11.59 ± 1.14 * | 5.24 ± 0.38 | 8.25 ± 0.73 * |
| 17–18 | 13.12 ± 0.77 | 14.06 ± 0.92 | 15.16 ± 1.03 | 17.12 ± 0.77 * | 11.99 ± 0.85 | 11.95 ± 1.19 * | 5.48 ± 0.58 | 7.6 ± 0.77 |
| 19–20 | 12.62 ± 0.45 | 12.75 ± 0.92 | 14.75 ± 0.52 | 16.57 ± 0.94 | 11.09 ± 0.7 | 10.48 ± 1.23 | 6.32 ± 0.6 | 8.44 ± 1.15 |

**B. ARV mmHg ±SEM (Inactive)**

| Days | MAP Control | MAP BPV | SBP Control | SBP BPV | DBP Control | DBP BPV | PP Control | PP BPV |
|---|---|---|---|---|---|---|---|---|
| Baseline | 12.09 ± 1.38 | 11.29 ± 0.54 | 13.53 ± 1.5 | 13.28 ± 0.7 | 11.27 ± 1.4 | 9.48 ± 0.45 | 4.74 ± 0.47 | 5.04 ± 0.45 |
| 1–2 | 12.15 ± 1.04 | 11.4 ± 0.69 | 14.08 ± 1.03 | 14.69 ± 0.77 | 10.95 ± 1.15 | 9.76 ± 0.68 | 4.92 ± 0.24 | 7.09 ± 0.55 |
| 3–4 | 12.23 ± 0.79 | 13.38 ± 0.33 * | 14.02 ± 0.76 | 17.27 ± 0.48 *# | 11.19 ± 1.02 | 11.29 ± 0.47 * | 4.76 ± 0.32 * | 7.9 ± 0.55 *# |
| 5–6 | 12.16 ± 0.43 | 13.63 ± 0.78 * | 14.2 ± 0.52 | 17.57 ± 0.78 *# | 10.68 ± 0.53 | 11.34 ± 0.86 * | 4.95 ± 0.3 | 8.23 ± 0.69 *# |
| 7–8 | 11.95 ± 0.49 | 13.73 ± 0.5 * | 13.64 ± 0.54 | 17.44 ± 0.53 *# | 10.88 ± 0.59 | 11.61 ± 0.67 * | 4.9 ± 0.32 * | 8.45 ± 0.89 |
| 9–10 | 12.9 ± 1.28 | 14.77 ± 0.66 * | 14.71 ± 1.49 | 18.52 ± 0.72 * | 11.92 ± 1.15 | 12.59 ± 0.64 * | 5.33 ± 0.31 * | 8.16 ± 0.61 *# |
| 11–12 | 12.49 ± 0.79 | 14.53 ± 0.57 * | 14.09 ± 0.82 | 18.65 ± 0.62 *# | 11.5 ± 1.07 | 12.09 ± 0.74 * | 5.24 ± 0.31 * | 8.5 ± 0.88 |
| 13–14 | 13.03 ± 0.75 | 13.96 ± 0.74 * | 14.85 ± 0.82 | 17.65 ± 0.53 * | 11.94 ± 0.99 | 11.56 ± 0.92 | 5.1 ± 0.38 * | 8.16 ± 0.74 *# |
| 15–16 | 13.35 ± 0.79 | 15.04 ± 0.87 * | 14.87 ± 0.93 | 18.85 ± 0.87 * | 12.21 ± 0.98 | 12.66 ± 1.07 | 4.97 ± 0.36 | 8.54 ± 1.06 * |
| 17–18 | 12.99 ± 0.7 | 15.61 ± 1.36 * | 14.19 ± 0.82 | 19.39 ± 1.09 *# | 11.47 ± 0.83 | 13.18 ± 1.67 | 5.02 ± 0.4 | 8.46 ± 1.03 |
| 19–20 | 12.89 ± 0.57 | 14.72 ± 1 * | 15.13 ± 0.74 | 18.96 ± 0.96 * | 11.36 ± 0.65 | 12.19 ± 1.25 | 6.21 ± 0.4 | 9.28 ± 1.33 |

ARV=average real variability, BPV=blood pressure variability, DBP=diastolic blood pressure, MAP=mean arterial pressure, PP=pulse pressure, SBP=systolic blood pressure.

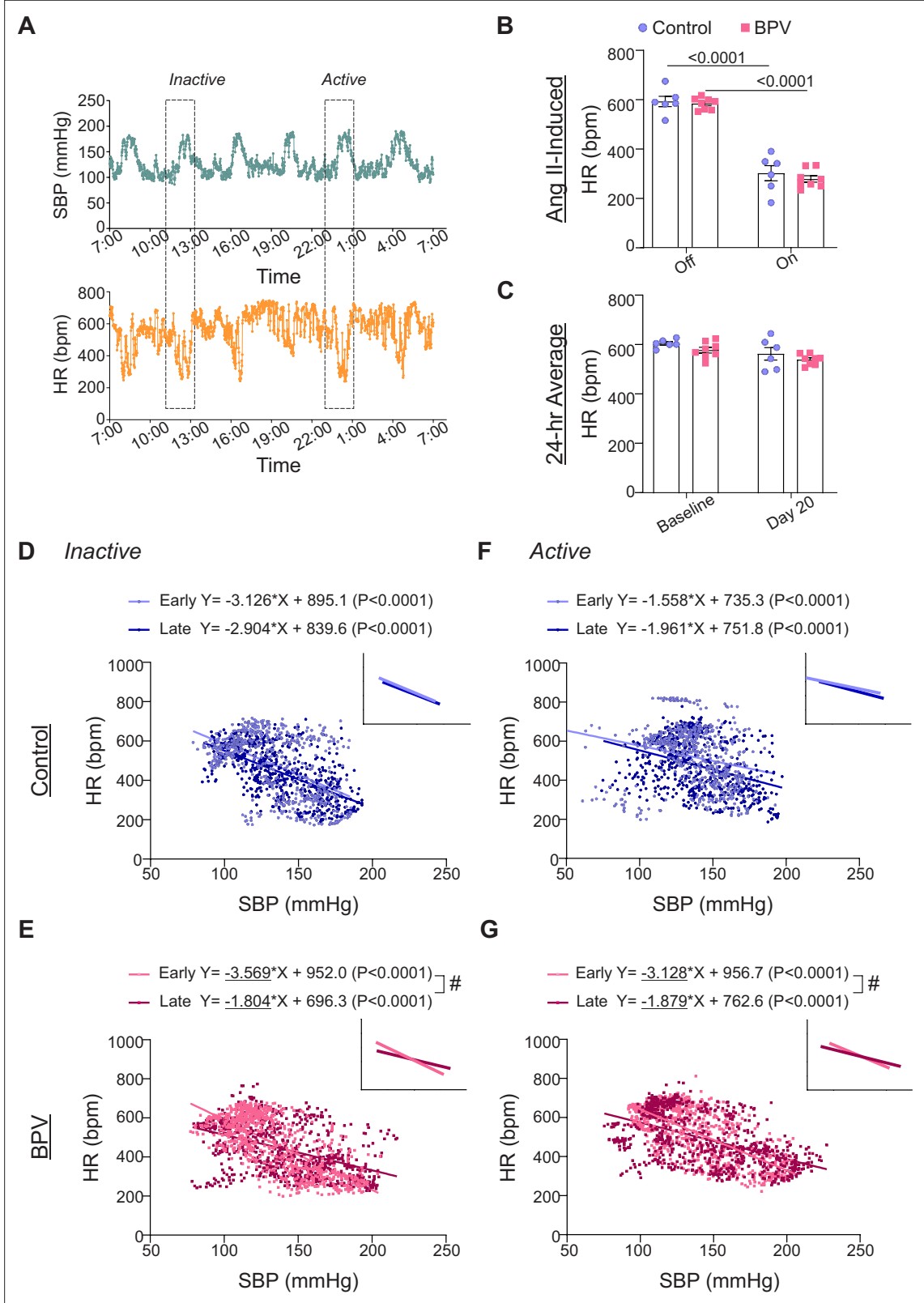

**Figure 3.** Chronic BPV suppresses bradycardic reflex. (**A**) Representative raw traces of 24 hr SBP (top) and HR (bottom) during intermittent Ang II infusions. (**B**) Five-minute average HR when the infusion pump is *Off* or *On* (actively infusing Ang II) for control and BPV groups, extracted from the inactive (daytime) period. (**C**) Twenty-four-hour average HR during the baseline period (~5 day saline infusion), and on day 20 of treatment (Day 20) for control and BPV mice. (**D**) Scatter plot of SBP and HR during Ang II infusion (100 min: 20 min before and after the 60 min pulse) in the inactive period for

*Figure 3 continued on next page*

*Figure 3 continued*

control and BPV mice (**E**). Linear trend lines for data shown in the insert. Early and Late periods correspond to days 3–5 and days 23–25 of the treatment phase, respectively. (**F**) Scatter plot of SBP and HR during the active period in controls and in BPV mice (**G**). Two-way ANOVA repeated measures followed by Sidak's multiple comparisons test (**B-C**, n=6 control mice, n=8 BPV mice). Simple linear regression and '#' denotes p<0.05 within group (**D**, **F**, n=6 control mice) (**E**, **G**, n=8 BPV mice). bpm = beats per minute, BPV = blood pressure variability, HR = heart rate, SBP = systolic blood pressure.

The online version of this article includes the following figure supplement(s) for figure 3:

**Figure supplement 1.** Transient Ang II infusions induced pulsatile blood pressure in control mice.

changes (p=0.0118), *Figure 5H*. However, because BPV mice exhibited lower baseline BP, although not significantly different from controls (p=0.18, *Figure 5D*), we examined whether differences in pressure range (i.e. the change in pressure between low and high BP intervals) contributed to the steeper slopes observed in BPV mice. *Figure 5I* illustrates the relationship between the lowest BP recorded during the acquisition period and the slope derived from the MAP-diameter linear regression analysis. Dashed lines represent the percent change in MAP relative to the minimal averaged BP recorded at the start of the imaging session. BPV mice showed a trend for greater percent change in BP (%△61±5.4 mmHg vs %△47±4.0 mmHg for control mice, p=0.06; *Figure 5I*, insert). However, the average BP range in control mice was still sufficient for establishing a correlation between BP and diameter, supporting the interpretation that differences in myogenic response stem from intrinsic vascular reactivity rather than limitations in the BP range.

The static cerebral autoregulation curve (CA), originally described by Lassen in 1959, is characterized by a plateau phase in the MAP-CBF relationship, sandwiched between lower and upper limit ranges (*Longden et al., 2017*). However, the shape of the curve has been challenged (*Lecrux et al., 2011*), with evidence suggesting a narrower plateau range and potential dilatory phases as BP deviates from this range (*McGrath et al., 2017*). To investigate pressure-diameter relationships in our cohort of mice, we analyzed parenchymal arterioles from control and BPV mice. As shown in *Figure 5J*, arterioles of both groups exhibited significant increases in tone (p<0.0001) as pressure rose from ~50–125 mmHg. When slopes from segmented regression analyses were compared across autoregulatory limits and plateau ranges (*McGrath et al., 2017*), our data revealed a significant decrease in arteriole diameter (p=0.0015) in control mice at pressures <50 mmHg (*Figure 5K*). Within the putative CA plateau range (51–100 mmHg) (*McGrath et al., 2017*), arterioles from both groups showed significant constrictions (p=0.01 for controls; p<0.0001 for BPV mice, *Figure 5L*), with steeper slopes observed in BPV arterioles (p<0.0001). At pressures >100 mmHg, BPV arterioles exhibited significant arterial dilations (p<0.001), whereas control arterioles maintained constrictions (p<0.001, *Figure 5M*). Together, these data suggest a leftward shift in the CA curve of BPV mice, characterized by enhanced myogenic constrictions at comparable pressures and significant arterial dilations at higher pressures where controls arterioles maintain tone, *Figure 5N*.

Previous studies have shown directional sensitivity in parenchymal arteriole responses, whereby increases in MAP evoke more efficient responses as compared to decreases in MAP. Thus, we separated and quantified diameter changes evoked by BP transitions from Low-to-High (increase MAP) and High-to-Low BP (decrease MAP), *Figure 5—figure supplement 1A–F*. Ang II infusion evoked similar increases in MAP when BP transitioned from Low-to-High Δ27±3.6 mmHg in controls and Δ25±3.6 mmHg in BPV mice, *Figure 5—figure supplement 1A*. In addition, MAP decreases were comparable between groups (-Δ21±2.8 mmHg for controls and -Δ25±3.1 mmHg for BPV mice) upon cessation of pump infusion (High-to-Low), *Figure 5—figure supplement 1D*. The significantly steeper negative slopes in BPV arterioles relative to controls (p=0.0117, Low-to-High) (*Figure 5—figure supplement 1C*) demonstrate increased parenchymal arteriole constriction to increases in MAP. Conversely, we also observed steeper negative slopes in BPV arterioles compared to controls (p=0.0044, High-to-Low) (*Figure 5—figure supplement 1F*) with decreases in MAP suggestive of enhanced arteriole dilations. Collectively, these findings indicate that chronic BPV heightens the responsiveness of parenchymal arterioles to acute BP fluctuations.

## Impaired NVC in high BPV mice

Because mechanisms underlying baseline CBF regulation differ from those evoked during the functional hyperemia response, we assessed whether chronic BPV affected functional-evoked outputs or neurovascular coupling (NVC). Functional hyperemia was assessed using two-photon imaging and

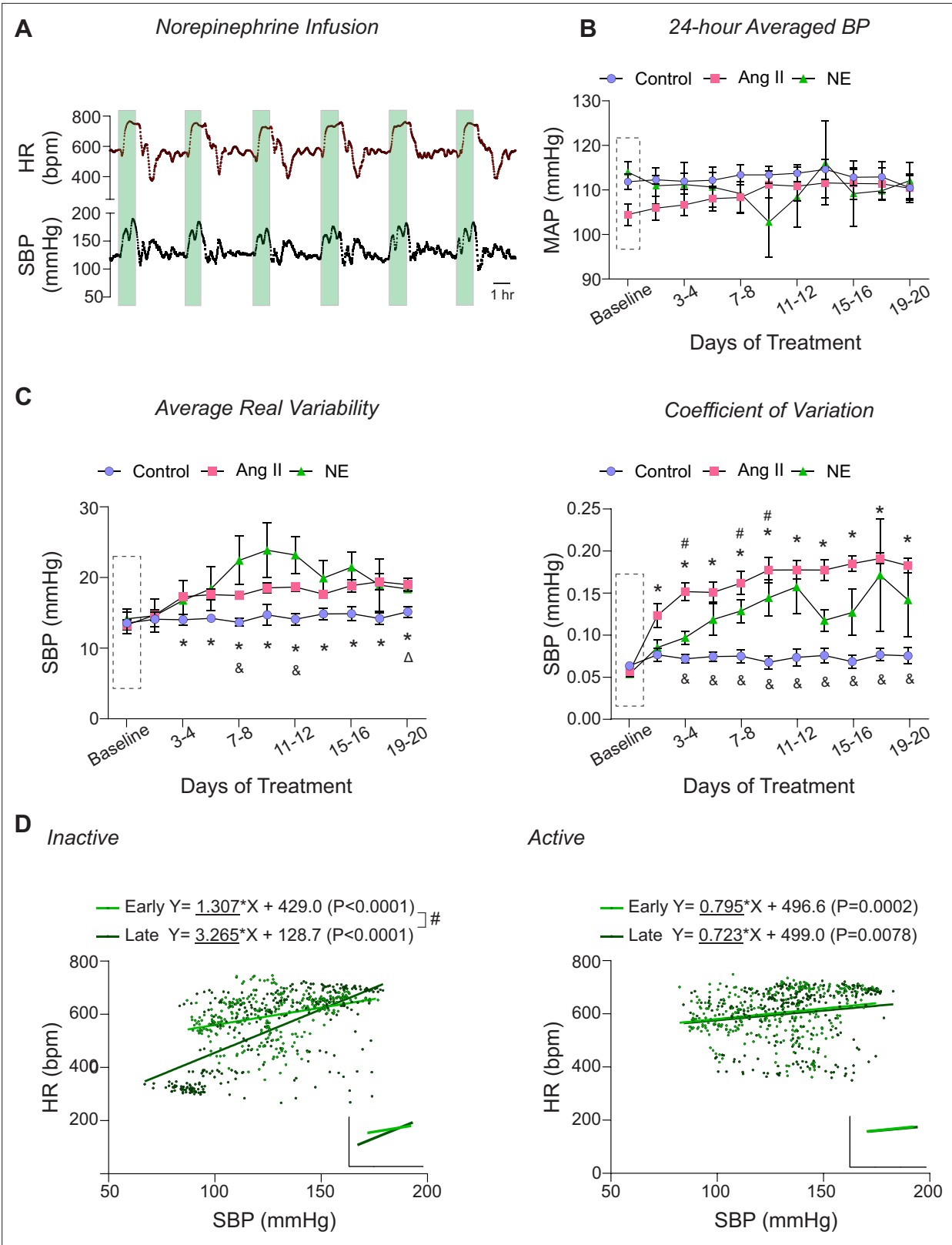

**Figure 4.** Norepinephrine infusion effects. (**A**) Representative raw trace of minute-to-minute 24 hr HR (top) and SBP (bottom). The green-shaded region indicates the 1 hr period when the pump is actively infusing NE (45 µg/kg/min). (**B**) Summary data of two-day average MAP during the baseline phase (Dashed rectangle, ~5 day saline infusion) and over 20 days of treatment. (**C**) Summary data of calculated two-day ARV of SBP (left) and CV of SBP (right). (**D**) Scatter plot of SBP and HR during NE infusion (100 min window: 20 min before and after the 60 min pulse) in the inactive (left) and active

*Figure 4 continued on next page*

*Figure 4 continued*

(right) periods, with linear trend lines shown in the insert. Early and Late periods correspond to days 3–5 and days 16–20 of the treatment phase. '*' and '#' denote within group comparisons (p<0.05 vs Baseline) for Ang II- and NE-infused mice, respectively. '&' and 'Δ' denotes between group comparisons (p<0.05 vs control group) for Ang II- and NE-infused mice, respectively. Two-way ANOVA repeated measures followed by Sidak's multiple comparisons test (B-C, n=6 mice). Simple linear regression (D, n=3 mice). Ang II = Angiotensin II, bpm = beats per minute, HR = heart rate, NE = norepinephrine, SBP = systolic blood pressure.

evoked using air puffs to stimulate mouse whiskers. Blood pressure measurements were simultaneously measured via telemetry, *Figure 6A*. While sensory-evoked vascular responses are commonly used as a proxy for neuronal activation and function (*Faraco et al., 2016*; *Lecrux et al., 2011*), few studies have considered (or reported) BP during these in vivo protocols. Thus, we asked if chronic and acute BP fluctuations alter the NVC response at the level of parenchymal arterioles, *Figure 6B*. NVC responses were compared during low and high BP periods corresponding to pumps being *off* or *on*, respectively, *Figure 6E*.

The average baseline (low BP) MAP was 78±2.5 mmHg and 71.0±3.4 for the control and BPV group, respectively, *Figure 6C*. Imaging was conducted at similar cortical depths within ~layers 1/2 border of the somatosensory cortex, corresponding to 107±1.0 μm and 96±0.4 μm for control and BPV arterioles, respectively, *Figure 6D*. To compare NVC responses at low and high BP, the pump reservoir for the control group was switched from saline to Ang II. Ang II infusion evoked significant (and comparable) increases in MAP, corresponding to 92±3.5 mmHg for controls and 92±5.7 mmHg for BPV mice, *Figure 6C*. *Figure 6E*, show a representative trace of the simultaneous MAP and arteriole diameter changes; WS denotes the stimulation period and 'a' and 'b' the timepoint from where diameters where measured before and after the stimulus, respectively and summarized in *Figure 6F*. Whisker stimulation caused significant dilations (vs. baseline) in all groups, regardless of MAP, *Figure 6G and H*. However, at higher BP, the magnitude of the NVC response in control mice was greater (p<0.0001, high BP vs low BP). Notably, this pressure-dependent response was abrogated (p=0.60) in the BPV group, *Figure 6G and H*. A closer look at the post-stimulus recovery phase showed a faster recovery rate (k=–0.45) for parenchymal arterioles of BPV mice during high BP periods compared to controls (p<0.0001), *Figure 6I*. These data revealed an acute pressure-dependent effect in control mice, with higher pressures resulting in greater sensory-evoked dilations. Additionally, the data shows that chronic BP fluctuations abrogated the pressure-dependent effects observed in control mice.

## Cognitive decline in high BPV mice

Given impaired neurovascular outputs and the established association between high BPV and cognitive decline (*McGrath et al., 2017*), we used the NOR and Y-maze tests to assess cognitive performance. Mice were subjected to behavioral testing during the saline-infused period and following 25 days of BPV (via Ang II infusions). Using NOR, a significant decrease in the recognition index (p=0.006) and discriminatory index (p=0.006) was observed (*Figure 7A*), supporting impaired recognition memory. No differences were observed in the short-term spatial working memory assessed via Y-maze alternation (p=0.25), *Figure 7B*.

Mice activity levels were evaluated as another aspect of behavior. Throughout the study, activity remained consistently higher during the active vs inactive periods (p=0.049 for control, p=0.0287 for BPV), *Figure 7C*. However, following 23–25 days of treatment (Late), BPV mice exhibited a significant reduction in activity when the pump was *on*. This decline was evident in the 24 hr average data (p=0.0111, *Figure 7D*), as well as during the active period of the 12 hr activity averages (pump *on* vs *off*, p=0.0122, *Figure 7F*). These data support that mice subjected to chronic BPV become less active when the BP increases or the pump status is *on*. Notably, while the infusion protocol preserves BP-related circadian rhythms, 25 days of chronic BPV negatively impacts both recognition memory and overall activity levels.

## Discussion

BPV has emerged as a risk factor for cognitive decline, yet its impact on brain function remains poorly understood. Here, we introduce a novel murine model of BPV, where pulsatile BP increases were induced via Ang II infusions without hypertension. Using in vivo two-photon imaging, we provide

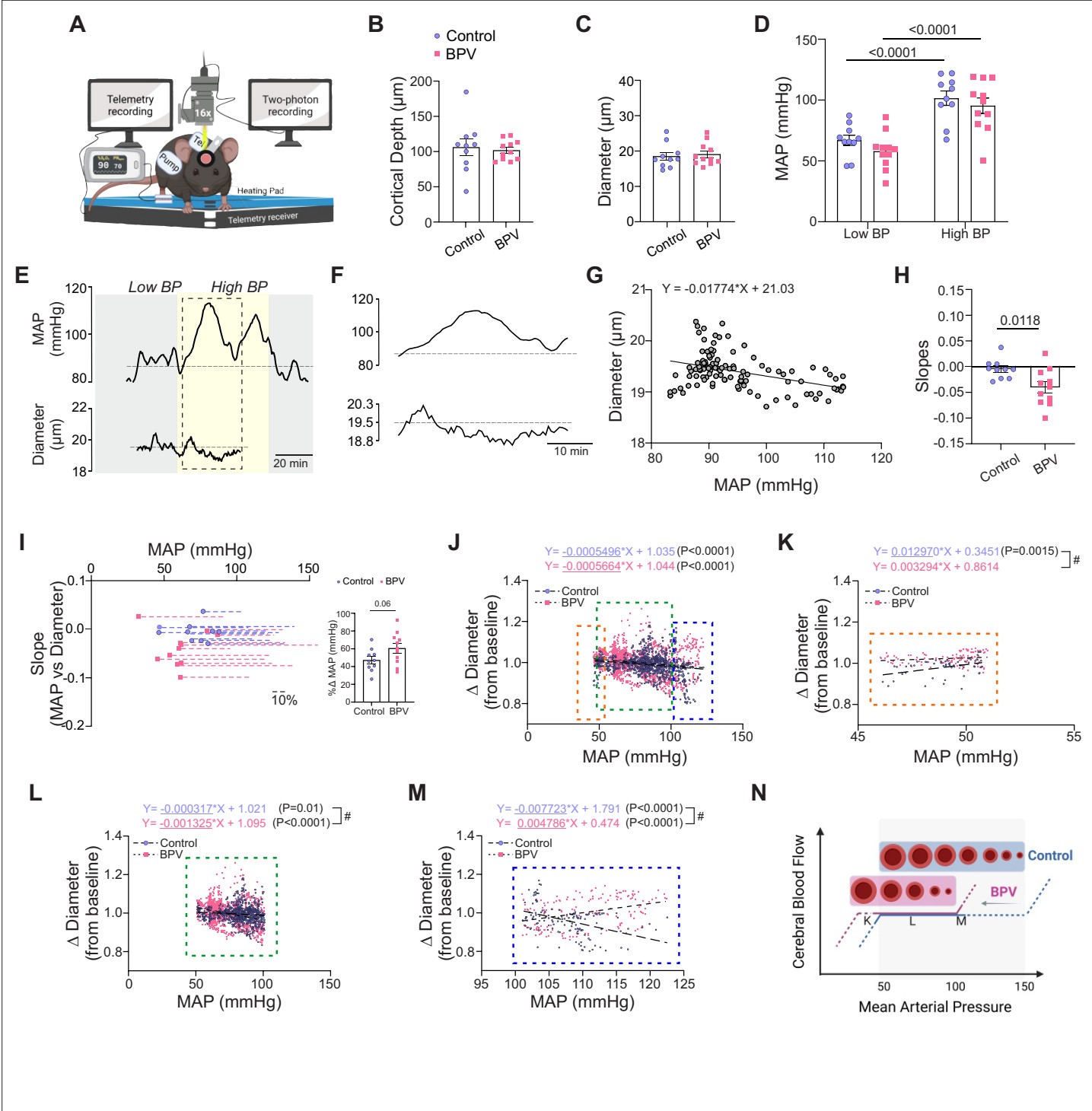

**Figure 5.** Enhanced myogenic responses in parenchymal arterioles of BPV mice. (**A**) Schematic of the experimental set-up, including a portable telemetry system for continuous blood pressure recordings alongside simultaneous 2 P imaging of parenchymal arteriole diameter. Created with BioRender.com. (**B**) Summary data for the average imaging depth below the brain surface. (**C**) Summary data for the average baseline diameters of imaged parenchymal arterioles. (**D**) Summary data of the average MAP recorded under infusion pump *off* conditions (low blood pressure, Low BP) and infusion pump *on* conditions (high blood pressure, High BP). (**E**) Representative raw trace (from a single mouse) showing MAP (top) and parenchymal arteriole diameter (bottom) over time. (**F**) Expanded data corresponding to dashed region in (**E**), highlighting Ang II-evoked BP changes. (**G**) Representative scatter plot and linear regression of MAP vs parenchymal arteriole diameter during an imaging session. (**H**) Summary data of MAP-diameter linear regression slopes. (**I**) Relationship between minimum MAP recorded during an imaging run and the slope shown in (**H**); dashed lines indicate the percent Δ change in MAP relative to its minimum value. The insert provides a summary of percent Δ change in MAP from the minimum

*Figure 5 continued on next page*

*Figure 5 continued*

MAP. (**J**) Scatter plot of MAP vs changes in arteriole diameter from baseline pressure (~70 mmHg), spanning a pressure range 46–122 mmHg. Data subsets extracted from (**J**): (**K**) MAP ≤51 mmHg (orange dashed region), (**L**) MAP 51-100 mmHg (green dashed region), and (**M**) MAP ≥101 mmHg (blue dashed region). (**N**) Illustration of the proposed leftward shift (and narrower plateau) in cerebral autoregulatory curve induced by chronic BPV. Created with BioRender.com. Unpaired t-test and Mann-Whitney test (**B-C, H-I**, n=10 runs/8 control mice, n=11 runs/8 BPV mice). Two-way ANOVA repeated measures followed by Sidak's multiple comparison test (D, n=10 runs/8 control mice, n=11 runs/8 BPV mice). Simple linear regression and '#' denotes p<0.05 between group (**J-M**, n=12 runs/9 control mice, n=8 runs/7 BPV mice). BPV = blood pressure variability, MAP = mean arterial pressure.

The online version of this article includes the following figure supplement(s) for figure 5:

**Figure supplement 1.** Directional myogenic responses.

**Figure supplement 2.** Exclusion criteria for myogenic response analysis.

evidence of altered vascular function in mice subjected to chronic BP fluctuations. In BPV mice, parenchymal arterioles (<30 μm in diameter) exhibited enhanced myogenic reactivity. In addition, chronic BPV impaired neurovascular outputs and cognitive function. Notably, the NVC response in control mice was pressure-dependent, with greater magnitudes observed when the arterial BP was increased–a relationship that was abolished in mice exposed to chronic BPV. These findings suggest that chronic BPV targets multiple physiological processes (e.g. bradycardic reflex, myogenic reactivity, neurovascular coupling) reinforcing its role as a critical risk factor for brain health.

Our goal was to induce rapid transient blood pressure pulses that significantly increased BPV, while maintaining a stable 24 hr average blood pressure. Ang II was selected as a pressor due to its potent vasoconstrictive properties and short half-life (*Al-Merani et al., 1978*). The subcutaneous dose was carefully titrated to induce BP elevations reaching physiological high levels (150–200 mmHg, SBP), achievable in non-diseased conditions and comparable to doses used in miniosmotic pumps (*Zimmerman et al., 2004*; *Nakagawa et al., 2020*; *Gonzalez-Villalobos et al., 2008*). The precise mechanisms underlying increased BPV remain unclear, though vascular dysfunction and arterial stiffness have been postulated as contributing factors (*Pucci et al., 2017*). Age is strongly correlated with arterial stiffness and isolated systolic hypertension (*Wallace et al., 2007*). Notably, SBP was modestly increased downstream of BPV in the current model, *Figure 1D*. In the BPV cohort, six out of eight mice exhibited an SBP increase of >Δ10 mmHg compared to baseline (*Figure 1D*), suggesting that elevated BPV may contribute to the early onset of systolic hypertension. Given that isolated systolic hypertension is a risk factor for overt hypertension (*Sagie et al., 1993*) and BPV precedes hypertension (*Özkan et al., 2022*), prolonged BPV beyond the 25 days treatment period could potentially lead to hypertension. Furthermore, we anticipate that sustained BPV and its resultant hypertension may drive cardiac remodeling, specifically, left ventricular hypertrophy. These findings underscore the need for future studies to investigate the long-term effects of BPV on cardiac morphology.

We explored norepinephrine as an alternative agent to induce BP variations, *Figure 4*. Similar to Ang II-infused mice, NE-treated mice exhibited a significant increase in BPV (i.e. increased CV) while maintaining minimal changes in the 24 hr averaged MAP. However, NE-induced BP pulses elicited a tachycardic response likely driven by NE stimulation of cardiac β1-adrenergic receptors (*Frishman, 2003*). Notably, chronic NE administration exaggerated this tachycardic response, with a steeper positive slope over time (Early vs Late; p<0.0001) during the Inactive period, *Figure 4D*. Additionally, NE-treated mice exhibited signs of lethargy, impaired mobility, tachypnea, and an increased rate of premature death beginning on day 11 of treatment. These findings emphasize the need for further investigation into alternative BPV pressors, as each may exert distinct off-target effects.

Our model underscores the importance of assessing BPV independently of average BP values. Retrospective studies in primary care patients reveal BPV occurrences in both hypertensive and non-hypertensive adults (*McAlister et al., 2021*), establishing BPV as a critical yet modifiable risk factor, often overlooked by standard in-office single BP measurements. In our model, the significantly higher 24 hr average PP in BPV mice suggests that PP is the most sensitive BP-related variable. Compared to controls, BPV mice exhibited persistently lower DBP and higher PP during the inactive period, indicative of a compensatory mechanism that maintained 24 hr MAP within physiological ranges. However, other factors mitigating hypertension onset cannot be ruled out.

The autonomic nervous system (ANS) tightly regulates BP via changes in heart rate, systemic vascular resistance, and stroke volume. Notably, the baroreflex plays a critical role in maintaining cerebral perfusion pressure (*Ogoh and Tarumi, 2019*). Acute BP elevations activate arterial baroreceptors,

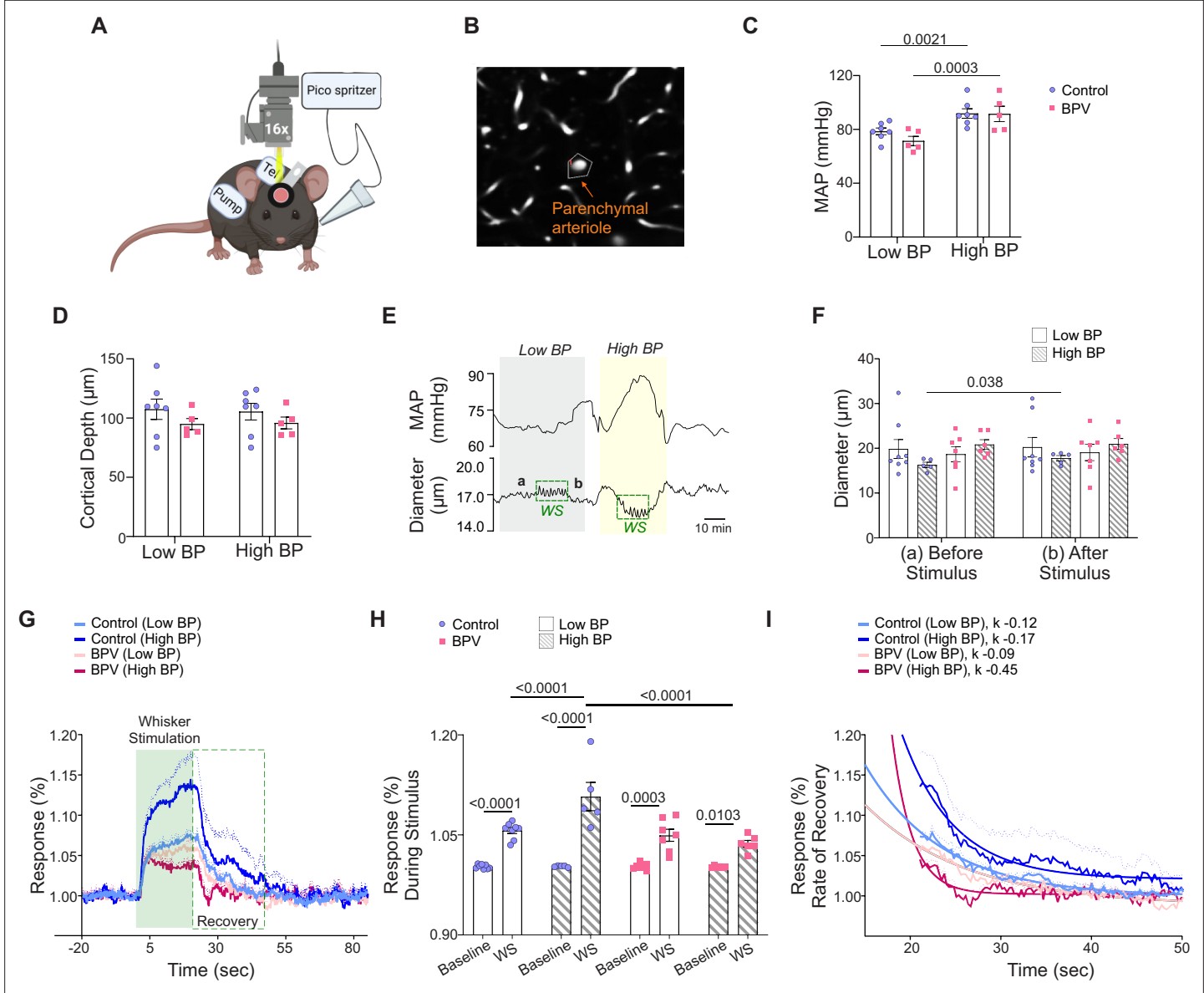

**Figure 6.** Suppressed neurovascular responses in parenchymal arterioles of BPV during low and high blood pressure periods. (**A**) Schematic of the experimental setup in which a picospritzer delivered a puff of air for whisker stimulation (WS) at 10 Hz for 20 s. Created with BioRender.com. (**B**) Representative image showing the mask (outlined) used to track changes in parenchymal arteriole diameter. (**C**) Summarized data of average MAP recorded during low blood pressure (Low BP; pump *off*) and high blood pressure (High BP; pump *on*) conditions. (**D**) Summary data of the average imaging depth below the brain surface. (**E**) Representative raw trace (from a single mouse) showing MAP (top) and parenchymal arteriole diameter (bottom) during low and high blood pressure. Dashed green squares indicate WS response, with 'a' denoting the 30 s pre-stimulus diameter and 'b' denoting the 30 s post-stimulus diameter, summarized in (**F**). (**G**) Normalized averaged arteriole diameter traces with corresponding error bars (dashed lines) during the WS response, shown as % change from baseline (20 s before stimulus), during the 20 sstimulus (green shaded region), and 64 s post-stimulus (green dashed square). (**H**) Summarized data of stimulus-induced arteriole responses (green shaded region in **G**). (**I**) Summary data of recovery time, with rate of decay (*k*) corresponding to 30 s post-stimulus period outlined in the green dashed square in (**G**). Two-way ANOVA repeated measures followed by Sidak's multiple comparisons test (**C**–**D**, n=7 control mice, 5 BPV mice) (**F**, n=5–8 control mice, n=6–7 BPV mice) and (**H**, n=5–8 control mice, n=6–7 BPV mice). One-phase exponential decay nonlinear fit (**I**, n=5–8 control mice, n=6–7 BPV mice). BPV = blood pressure variability, *k*=rate of decay, MAP = mean arterial pressure, WS = whisker stimulation.

The online version of this article includes the following figure supplement(s) for figure 6:

**Figure supplement 1.** Seasonal effects on mean arterial pressure.

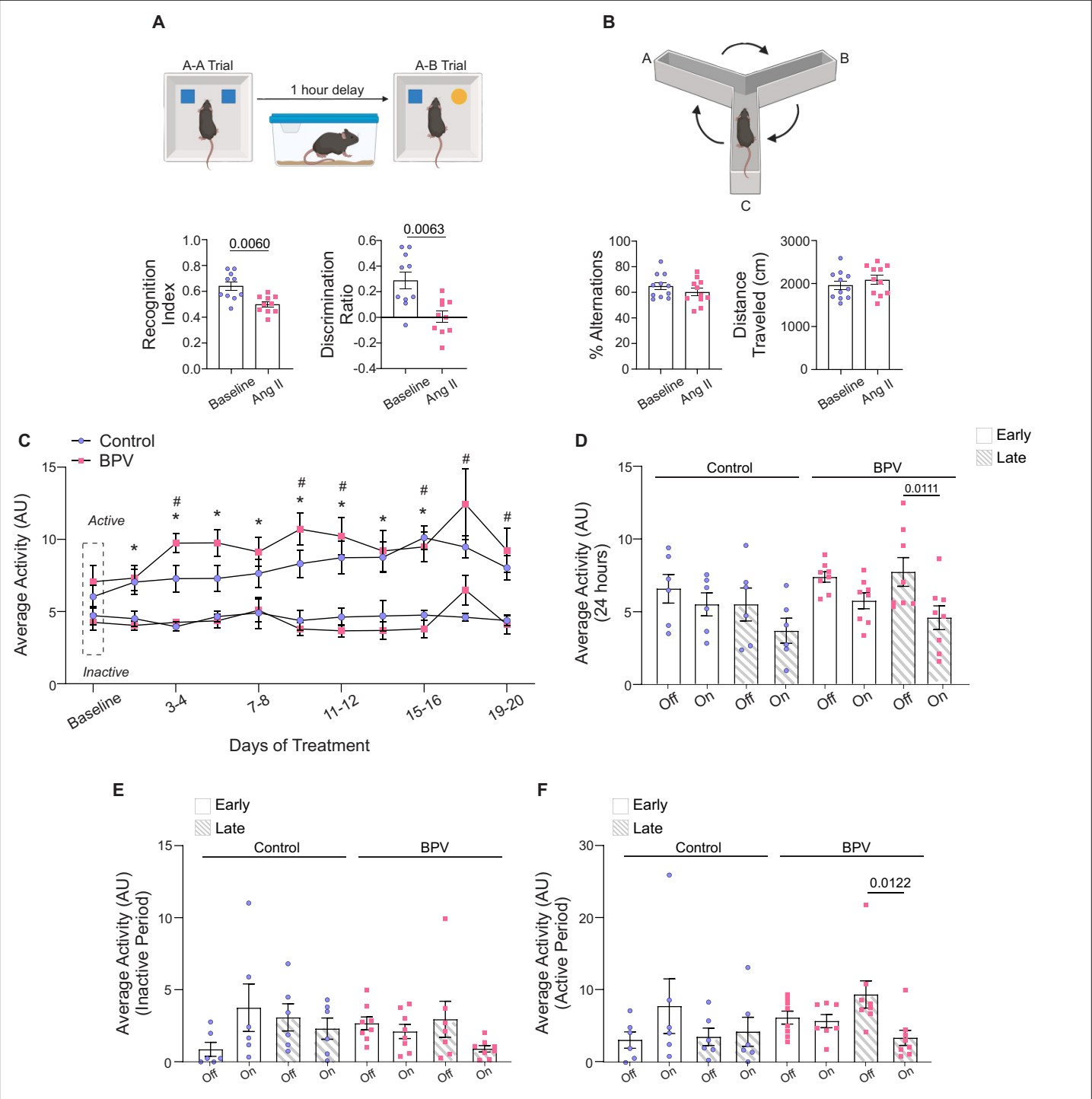

**Figure 7.** Behavior and altered cognitive function of BPV mice. (**A**) Schematic of the novel object recognition test with a 1 hr delay between the A-A and A-B trials (top), accompanied by summarized recognition and discriminatory index data for mice during the saline infusion period (baseline) and following 25 days of pulsatile Ang II infusion (Ang II). Created with BioRender.com. (**B**) Diagram of 10 min spontaneous Y-maze experimental setup (top) and the summary of % alternation and distance traveled during baseline and after 25 days of Ang II treatment (bottom). (**C**) Summary data of two-day, 24 hr activity averages throughout treatment during active and inactive periods, with the Baseline phase (~5 day saline infusion) marked by the dashed rectangle. (**D**) Summary data of 24 hour averaged activity recorded when the infusion pump was *on* vs *off* during Early (days 3–5) and Late (days 23–25) treatment phases. (**E**) Summary of activity when the pump is *on* vs *off* during the Early and Late phases of treatment for the inactive cycle. (**F**) Summary of activity when the pump is *on* vs *off* during the Early and Late phases of treatment for the active cycle. Paired t-test (**A**, n=10 BPV mice), (**B**, n=11 BPV mice). Two-way ANOVA repeated measures followed by Tukey's comparison test (**C**, n=13 control mice, n=8 BPV mice) (**D-F**, n=6 control mice, n=8 BPV

*Figure 7 continued on next page*

*Figure 7 continued*

mice). '*' denotes p<0.05 vs Baseline for control, active period. '$' and 'Δ' denote p<0.05 (active vs inactive period) for control and BPV, respectively. AU = arbitrary units, BPV = blood pressure variability.

which inhibit sympathetic outflow (via enhanced vagal activity) and reduce HR and peripheral vascular resistance (*Ogoh and Tarumi, 2019*), thereby lowering arterial BP. However, to sustain relatively stable CBF, baroreflex-mediated inhibition of sympathetic outflow and arteriole dilation is counterbalanced by the intrinsic myogenic properties of the cerebral vasculature, which constricts in response to increases in arterial BP. Dysregulation of these dynamic processes heightens the risk of both cerebral hypoperfusion and hyperperfusion. Notably, impaired baroreflex sensitivity has been linked to aging (*Teixeira et al., 2019*) and mild cognitive impairments (*Tarumi et al., 2015*).

In our model, chronic BPV significantly blunted the bradycardic response, suggestive of ANS dysregulation, potentially driven by enhanced sympathetic activity, reduced parasympathetic activity, or a combination of both. Previous studies indicate that chronic Ang II infusion suppresses and resets baroreceptor sensitivity independently of the pressure effect, with partial reversal occurring within 30 min post-Ang II infusion (*Brooks, 1995*). Unlike conventional sustained Ang II infusion approaches (osmotic pump models), our protocol delivers Ang II intermittently at 3- or 4 hr intervals. This regimen likely mitigates desensitization and Ang II receptor internalization, a response well-documented in sustained infusion models (*Hunyady et al., 2000*; *Guo et al., 2001*). Furthermore, the same Ang II dose evoked comparable BP increases at the start and end of treatment. Thus, our results support diminished baroreflex responses, aligning with previous reports linking elevated BPV to autonomic dysregulation (*Zhang et al., 2012*), though the underlying cellular mechanisms remain poorly understood.

Cerebral autoregulation serves as a protective mechanism against fluctuations in cerebral perfusion pressure (*Iadecola and Davisson, 2008*). Vessels upstream from cerebral capillaries, including pial arteries (*Klein et al., 2022*; *Fog, 1938*; *Lassen, 1959*) and larger extracranial vessels (*Kontos et al., 1978*; *Faraci et al., 1987*; *Faraci and Heistad, 1990*), help buffer arterial pressure changes, thereby limiting excessive variations in CBF. In response to chronic and sustained hypertension, parenchymal arterioles exhibit enhanced reactivity to pressure increases (*Pires et al., 2015*; *Iddings et al., 2015*; *Diaz et al., 2019*). This is further accompanied by arteriole remodeling (*Pires et al., 2015*; *Diaz-Otero et al., 2017*; *Rigsby et al., 2011*). Although our model does not involve sustained hypertension, we present in vivo evidence of heightened myogenic responses of parenchymal arterioles to both BP elevations and reductions. These observations are particularly intriguing as they suggest potential adaptive mechanisms in response to large blood pressure fluctuations.

Chronic BP elevations may drive upregulation of voltage-dependent calcium channels (VDCC), or suppression of $K^+$ currents (*Koide et al., 2021*). *Cipolla et al., 2014*, reported that parenchymal arterioles are approximately 30-fold more sensitive to VDCC inhibition via nifedipine than middle cerebral arteries. Notably, VDCC expression increases in hypertension (*Lozinskaya and Cox, 1997*; *Simard et al., 1998*; *Pratt et al., 2002*; *Pesic et al., 2004*; *Sonkusare et al., 2006*; *Nieves-Cintrón et al., 2018*; *Navedo et al., 2010*), and long-acting calcium channel blockers (i.e. amlodipine), are among the most effective treatments for BPV (*de la Sierra, 2023*; *Levi-Marpillat et al., 2014*). In an effort to mitigate excessive vasoconstriction and optimize blood flow, during periods of BP-induced constrictions, vasodilatory signals may accumulate. Consequently, when BP decreases, parenchymal arterioles in BPV mice may exhibit greater dilation compared to controls, *Figure 5—figure supplement 1F*. Augmented vasodilation has been reported as adaptive response in rodent models of chronic hypoperfusion (*Kim et al., 2021*; *Chan et al., 2019*). The dynamic nature of BP fluctuations in our model (i.e. increases and decreases) uniquely allows observations of multiple processes occurring concurrently (e.g. enhanced vasoconstrictions in response to high BP, and enhanced dilations potentially resulting from vasoconstriction-evoked ischemia).

If this interpretation holds, our model may reflect conditions in which parenchymal arterioles experience both high pressure and ischemia. The resulting heightened myogenic responses may amplify diameter changes, potentially diminishing pressure buffering capacity and increasing vulnerability to extreme BP fluctuations, predisposing brain tissue to episodic ischemia and/or hyperperfusion. Future studies addressing the impact of BPV on capillary perfusion may provide further insights into the link between BPV and brain pathology, including microbleeds (*Zhang et al., 2024*; *Ma et al., 2020*).

Chronic BPV significantly blunted the NVC response to whisker stimulation. In control mice, NVC exhibited a pressure-dependent effect, with greater response magnitudes observed at higher arterial BP levels—an effect that was abolished in BPV mice. One possible explanation is that the enhanced NVC response (under high BP conditions) in control mice stems from greater baseline arteriole tone, facilitating a more efficient dilatory response to vasoactive signals released during sensory-evoked stimuli. Conversely, the heightened reactivity of parenchymal arterioles to intravascular pressure in BPV mice (*Figure 5H*) suggests a predominant vasoconstrictive state (*Thorin-Trescases and Bevan, 1998*), potentially accompanied by reduced availability of vasodilatory mediators involved in NVC (e.g. nitric oxide, epoxyeicosatrienoic acids (EETs), cyclooxygenase-derived prostaglandins; *Tarantini et al., 2015*; *Ma et al., 1996*). Thus, the blunted NVC response observed during elevated BP periods in BPV mice may also result from impaired vasodilation.

Additionally, upon termination of WS, vasoconstrictive mechanisms—exacerbated by elevated BP are unmasked, leading to a faster recovery in arteriole diameter post-stimulus (*Lim et al., 2024*). Another potential explanation is a neuronal mechanism, where diminished neural activity or reduced release of vasodilatory signals may contribute to the impaired NVC response. Further studies should investigate the impact of high BPV on the neurovascular complex under awake conditions, recognizing that factors such as activity and arousal (*Tran et al., 2018*; *Renden et al., 2024*), can elevate BP and potentially confound pressure-dependent neurovascular effects.

Beyond blunting NVC, our study demonstrates that chronic BPV contributes to cognitive decline, as evidenced by decreased discrimination and recognition indices in the NOR test. Clinical studies have similarly linked elevated BPV to impairments across multiple cognitive domains, including memory and executive function (*Epstein et al., 2013*; *Sible and Nation, 2023*). Post hoc analyses of the SPRINT MIND trial revealed that patients with elevated BPV experienced the most rapid decline in processing speed (*Sible and Nation, 2023*). Additionally, findings from the Alzheimer's Disease Neuroimaging Initiative indicated that greater SBP variability correlated with poorer episodic memory performance (*Epstein et al., 2013*).

To assess cognitive function, we employed the NOR test to evaluate episodic memory (*Antunes and Biala, 2012*) and the spontaneous Y maze test to assess short-term spatial memory (*Kraeuter et al., 2019*). Because telemetry systems necessitated individual housing to avoid signal interference, all mice used in this study were single-housed. Previous studies on C57BL/6 mice have reported conflicting effects of single housing on learning and memory (*Liu et al., 2020*; *Benfato et al., 2022*; *Magalhães et al., 2024*), with some suggesting stress-induced deficits and others reporting no impact (*Hu et al., 2023*; *Panossian et al., 2020*; *Lander et al., 2017*; *Smolensky et al., 2024*). These discrepancies may stem from variables such as age at isolation onset, duration of isolation, and behavioral test conditions. To minimize confounding factors, we implemented stringent efforts to ensure consistency across all groups, including standardized cage changes, food replenishment, and pump refills conducted on a single day between 1:00 and 4:00 PM. Our results indicate that while 25 days of chronic BPV does not affect short-term spatial memory, it significantly impairs episodic memory.

## Conclusions

Our study demonstrates that high BPV disrupts cellular communication in the neurovascular complex, contributing to cognitive decline. Importantly, our model elicited substantial BP fluctuations while preventing the onset of hypertension, providing direct evidence of BPV's causal role in impaired neurovascular function. These findings support the inclusion of BPV assessment as a critical screening and diagnostic tool alongside conventional BP measurements to address cardiovascular and neurovascular risks. Moreover, the adaptable murine model developed in this study offers a robust framework for future investigations into BPV's impact on brain function.

## Methods

### Animals

All experiments were conducted in middle-aged (12–15 month-old) male C57BL/6 mice (Jackson Laboratories) under protocols approved by the Institutional Animal Care and Use Committee of Augusta University (AU). Mice were housed at 20–22°C under a 12 hr:12 hr light-dark cycle with ad libitum access to food and water. Female mice were not included in the present study due to high

post-surgery mortality observed in 12–14 month-old mice following complex procedures. To minimize confounding effects of differential survival and to establish foundational data for this model, we restricted the investigation to male mice.

## Craniotomy surgery for chronic window

Surgeries were conducted using the aseptic technique. Animals were injected with dexamethasone and meloxicam 2–4 hr before surgery to prevent edema and/or inflammation. General anesthesia was induced with isoflurane (2%) and following loss of reflex, hair was removed from the scalp and the mouse was transferred to a sterile field. A single injection of ketamine/dexdomitor (60 mg/kg/0.5 mg/kg) was administered to maintain anesthesia in the sterile field. The scalp was scrubbed with betadine/alcohol (three times each, alternating). A scalp incision was made (~1 cm). A small aluminum head holder (300 mg) was fixed to the skull using cyanoacrylate glue followed by dental cement and a small craniotomy opened over the cortex. The bone flap was removed. A glass coverslip (3 mm) was glued to a 4 mm coverslip and placed into the craniotomy with the inner glass touching the cortex. The edges of the coverslip were then sealed with cyanoacrylate glue followed by dental cement covering everything, including the wound margins.

## Telemetry and iPrecio pump implantation surgery

Following a 3 week recovery period after craniotomy surgery, mice were implanted with a programmable pump (iPrecio, SMP-310R) and a biotelemetry transmitter device (PA-C10, Data Science International). Mice underwent brief anesthesia in a small chamber using 5% isoflurane for induction, followed by maintenance of anesthesia via a nose mask with 1.5–2% isoflurane. The neck fur was carefully clipped on both the anterior and posterior sides. Subsequently, mice were placed in a sterile field in a ventrally recumbent position, and a 1.5–2 cm horizontal incision was made. The pump and biotelemetry transmitter device were then implanted subcutaneously. The pump catheter was trimmed to 0.5 cm for subcutaneous drug delivery, while the telemeter catheter was tunneled subcutaneously over the shoulder, with its tip positioned within the left carotid artery. Closure of incisions was performed using 5–0 sutures, after which the mice were gradually brought out of anesthesia. It is important to note that due to the complexity of the surgeries involved, mice used for behavioral studies were not implanted with a telemetry catheter or cranial window. However, these mice were individually housed to simulate the conditions of the telemetry cohort.

## Blood pressure assessment and BPV induction

After recovery from surgery, mice were housed individually in standard mouse cages under the conditions described above and assigned to a control or experimental BPV group. During this recovery period (~7 days), both groups were infused with saline to determine baseline cardiovascular parameters. Blood pressure signals encompassing mean arterial pressure (MAP), systolic blood pressure (SBP), diastolic blood pressure (DBP), heart rate (HR), and pulse pressure (PP) (determined from the difference between the SBP and DBP) were continuously sampled at 250 Hz for 10 seconds and collected every 30 s, for 24 hr per day. Once a baseline MAP was established, the saline in the infusion pump (in the BPV group) was substituted with angiotensin II (Ang II, Sigma Aldrich, A9525) and administered intermittently at a calculated dose of 3.1 µg/hr (every 3–4 hr) for 25 days.

### Controls receiving Ang II

To facilitate between-group comparisons (control vs BPV), a cohort of control mice were subjected to the same pump infusion parameters as BPV mice but for shorter durations receiving Ang II infusions on days 3–5 and, then again, on days 21–25 of the experimental protocol. This paradigm was used for experiments assessing pressure-evoked responses, including bradycardic reflex, myogenic response, and functional hyperemia at high BP.

### BPV induction with norepinephrine

To validate the intermittent infusion protocol as an effective method for inducing significant BPV and not limited to Ang II, a separate cohort of mice received norepinephrine (NE, 45 µg/kg/min) using the same subcutaneous delivery protocol (2 µL every 4 hr).

## Two-photon imaging

On the day of two-photon imaging acquisition, a mouse was anesthetized with chlorprothixene (0.04 cc) and a low dose of isoflurane (≤0.8%); a protocol to mildly sedate mice (*O'Herron et al., 2022*). A retro-orbital injection of Texas red (70 kDa dextran, 5% [wt/vol] in saline, 40 µl) was administered to label blood vessels. The mouse was head-fixed and a picospritzer (positioned on the contralateral hemisphere intended for imaging) was used to deliver a puff of air for whisker stimulation (WS) at a rate of 10 Hz for 20 s repeated six to eight times with ~90 s delay between each WS during an imaging session. Imaging sessions were conducted 106±3.6 µm below the brain's surface while pump infusion was *off* (low BP period) and again during Ang II infusions (high BP period). Images were collected at a rate of 3.75 frames/second.

## Behavioral studies

Behavioral tasks were performed by an experimenter at the AU small animal behavioral core (SABC), who was blinded to the experimental protocol. BPV mice were subjected to behavioral tests at two time points. First, during the saline infusion period, with observations considered as baseline, and second, 25–26 days post-Ang II infusions. About 30 min before training and testing, animals were brought to the testing room to acclimate. Animals remained in the laboratory for 15 min following study completion. To eliminate olfactory cues, animal droppings were removed, and the space and objects were cleaned (dilute 50% (vol/vol) ethanol solution) between sessions where appropriate.

### Novel object recognition (NOR) task

Animal behavior was assessed using the previously described NOR task procedure (*Callahan et al., 2021*). Animals were acclimated and then familiarized with an opaque plastic chamber (78.7 cm × 39.4 cm × 31.7 cm) containing bedding for 10 min. The next day, a training session involved animals exploring two identical objects for 10 min before returning to their home cages. Object recognition was assessed one hour later by placing an animal in the NOR apparatus with a familiar object and a new or novel object for 10 min. The objects' positions and roles (familiar or novel) were randomly assigned. Objects were positioned about 40 cm apart, 19.3 cm from the two short walls and 19.3 cm from the two long walls of the chamber. Exploration of an object was defined as direct interaction with nostrils or head position towards the object at a distance ≤2 cm. For data inclusion, a mouse had to explore an object for at least 4 s and spend a minimum of 10 seconds of total object exploration. Recognition index was calculated using the formula: $Recognition\ Index = \frac{(time\ spent\ at\ novel\ object)}{(time\ spent\ at\ novel+familiar\ objects)}$. The discrimination index was calculated as: $Discrimination\ Ratio = \frac{(time\ spent\ at\ novel-familiar\ object)}{(time\ spent\ at\ novel+familiar\ objects)}$. The one-hour delay for testing was determined based on earlier NOR tests assessing the cohort's recognition capacity at 1, 4, and 24 hr delays. The animals exhibited normal recognition capacity with a 1 hr delay but failed at 4- and 24 hr delays. Hence, the one-hour delay was selected for subsequent tests.

### Y-maze

Y-maze tasks were performed 24 hr after NOR tasks. The Y-maze consisted of three arms (35.4 cm long, 9.9 cm wide with a height of 13.8 cm). Spontaneous alternation behavior was assessed by randomly placing mice in one of the three arms and allowed them to explore for 10 min. Arm entries were visually scored into a series where a triplet set of arm entries constituted an alternation. An alternation was defined as successive consecutive entries into three different arms, and maximum alternations as the total number of arm entries minus 2. Percent alternation was calculated as the proportion of true alternations out of maximum alternations ((# of true alternations/# of maximum alternations) × 100).

## Data analysis

Two-day averaged values of 24 hr or 12 hr (active and inactive period) BP variables were compared to a baseline phase corresponding to saline infusion (~5 days). BPV was calculated using the average real variability (ARV) index ($ARV = \frac{1}{N-1} \sum_{K=1}^{N-1} \times |BP_{K+1} - BP_{K}|$), where K is the order of measurements and N denotes the number of BP readings. Absolute BP values were extracted every 15 min and averaged for the hour. The coefficient of variation (CV) was also used to calculate BPV. CV was calculated as the ratio of the standard deviation of hourly BP averages to the 24 hr BP average

$(CV = \frac{\text{Standard Deviation (Hourly BP Average)}}{\text{24 Hour BP Average}})$. Similar to BP variables, two-day averages of 24 hr and 12 hr ARV and CV were compared to the average value of the last 5 days during the baseline phase. Parenchymal arteriole diameter responses to changes in MAP (myogenic responses) and whisker stimulation were analyzed using MatLab and ImageJ (*Chhatbar and Kara, 2013*; *O'Herron et al., 2016*). Simple linear regression determined from the relationship of SBP and HR assessed bradycardic responses; MAP and arteriole diameter changes assessed myogenic reactivity. For neurovascular coupling experiments, the baseline was defined as 30 frames before stimulus onset. The stimulus-response comprised 100 frames encompassing the 20 s stimulus and the first 6 s post-stimulus. Six to eight WS events were averaged for each run and responses were normalized to the baseline average.

## Exclusion criteria

Mice that died prematurely (before treatment onset) or exhibited abnormally elevated or low blood pressures following telemetry surgery, likely due to surgical complications, were excluded from the study.

During two-photon imaging sessions, we observed unexplained, random transient dilatory events (*Figure 5—figure supplement 2B, D*) occurring in both control and BPV mice across low- and high-BP periods. Because similar random dilations have been documented in other mouse strains under comparable in vivo imaging conditions, these events may be attributed to the effects of the sedative. To isolate pressure-evoked parenchymal arteriole responses in the myogenic studies (*Figure 5*), transient dilatory events associated with baseline tone were excluded from the analysis, *Figure 5—figure supplement 2F*.

## Study limitations

Experiments involving control and BPV mice were conducted across different seasons, raising the possibility for seasonal influences (*Suckow and Tirado-Muñiz, 2023*; *Kastenmayer et al., 2006*). To account for this, seven additional control mice were included to the original thirteen control mice, *Figure 6—figure supplement 1*. Six of these mice underwent identical treatment but were allocated to a separate branch of the study and did not receive chronic cranial window implantation—a procedure that does not significantly affect BP, *Figure 6—figure supplement 1A*. No differences in 24 hr averaged MAP were observed between control mice when grouped categorized by Georgia's seasonal climate Fall-Winter (October-April) vs Spring-Summer (May-September), *Figure 6—figure supplement 1B*. Given the absence of seasonal effects on BP and the fact that mice were sourced from two independent suppliers (Jackson Laboratory and the National Institute of Aging Aged Colonies), we attribute our observations primarily to treatment effects.

GraphPad Prism 10 software (GraphPad Software, La Jolla, CA) was used for all statistical analyses. Values are expressed as mean ± SEM. A minimum of three mice was used for each experimental data set, and the specific sample size (n) is defined in figure legends. Data was tested for normal distribution, and statistical tests were used accordingly. Differences between two means within groups were determined using paired Student's *t*-test. Differences between groups were determined using Student's unpaired *t*-test or two-way ANOVA with corresponding multiple comparison *post hoc* test specified in figure legends. Statistical significance was tested at a 95% ($p<0.05$) confidence level denoted with the corresponding symbol in figure legends.

# Acknowledgements

This research was supported by 1R56NS123644, 5R01NS123644 to JAF and 24POST1196351 AHA and 3R01NS123644-02S2 to PJM. We acknowledge the contributions of animals from the National Institutes of Aging. In addition, we thank Kathleen A Coleman, and Cameron Folk, for their technical support in conducting telemetry and pump-related surgeries. We also thank Dr Alvin Terry and Daniel Beck the AU Small Animal Behavioral Core for their assistance with the behavioral cognitive tests. Diagrams created with Biorender.com

## Additional information

### Funding

| Funder | Grant reference number | Author |
|---|---|---|
| National Institute of Neurological Disorders and Stroke | 5R01NS123644 | Jessica A Filosa |
| National Institute of Neurological Disorders and Stroke | 3R01NS123644-02S1 | Perenkita J Mendiola |
| National Institute of Neurological Disorders and Stroke | 1R56NS123644 | Jessica A Filosa |
| American Heart Association | 10.58275/aha.24post1196351.pc.gr.190837 | Perenkita J Mendiola |
| National Institute of Neurological Disorders and Stroke | 3R01NS123644-02S2 | Perenkita J Mendiola |

The funders had no role in study design, data collection and interpretation, or the decision to submit the work for publication.

### Author contributions

Perenkita J Mendiola, Conceptualization, Formal analysis, Methodology, Writing – original draft, Project administration, Writing – review and editing; Philip O'Herron, Software, Formal analysis, Methodology, Writing – review and editing; Kun Xie, Formal analysis, Supervision, Methodology; Michael W Brands, Conceptualization, Supervision, Methodology; Weston Bush, Methodology; Rachel E Patterson, Formal analysis, Methodology, Writing – review and editing; Valeria Di Stefano, Formal analysis, Methodology; Jessica A Filosa, Conceptualization, Data curation, Formal analysis, Supervision, Funding acquisition, Writing – original draft, Project administration, Writing – review and editing

### Author ORCIDs

Perenkita J Mendiola (ID) https://orcid.org/0000-0003-0428-6272
Philip O'Herron (ID) https://orcid.org/0000-0002-8137-9432
Jessica A Filosa (ID) https://orcid.org/0000-0001-7536-2039

Reviewer #1 (Public review): https://doi.org/10.7554/eLife.104082.3.sa1
Author response https://doi.org/10.7554/eLife.104082.3.sa2

## Additional files

### Supplementary files

Supplementary file 1. Pulsatile Ang II infusions induced an increase in blood pressure variability (BPV) measured by coefficient of variance. (A-B) Summary data of calculated two-day average coefficient of variance (CV) of MAP, SBP, DBP, and PP during the active period (A) and inactive period (B). CV was calculated as the ratio of the standard deviation of hourly BP averages to 24 hr mean blood pressure. Two-way ANOVA repeated measures followed by Dunnett's comparison test; n=13 control mice, n=8 BPV mice. '*' denotes $p<0.05$ vs Baseline and '#' denotes $p<0.05$ between groups. BPV = blood pressure variability, CV, coefficient of variance, DBP = diastolic blood pressure, MAP = mean arterial pressure, $P$=pulse pressure, SBP = systolic blood pressure.

MDAR checklist

Source data 1. Raw numerical data that are represented as a graph in a figure or as a summary table.

### Data availability

Raw data is in *Source data 1* and *Supplementary file 1* contains additional data.

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
