## [Editor Report · eLife Assessment]

This is an **important** study that demonstrates that blood pressure variability impairs myogenic tone and diminishes baroreceptor reflex. The study also provides evidence that blood pressure variability blunts functional hyperemia and contributes to cognitive decline. The evidence is **compelling** whereby the authors use appropriate and validated methodology in line with or more rigorous than the current state-of-the-art.

---

## [Referee Report · Reviewer #1 (Public review)]

This study examined the effect of blood pressure variability on brain microvascular function and cognitive performance. By implementing a model of blood pressure variability using intermittent infusion of AngII for 25 days, the authors examined different cardiovascular variables, cerebral blood flow and cognitive function during midlife (12-15-month-old mice). Key findings from this study demonstrate that blood pressure variability impairs baroreceptor reflex and impairs myogenic tone in brain arterioles, particularly at higher blood pressure. They also provide evidence that blood pressure variability blunts functional hyperemia and impairs cognitive function and activity. Simultaneous monitoring of cardiovascular parameters, in vivo imaging recordings, and the combination of physiological and behavioral studies reflect rigor in addressing the hypothesis. The experiments are well designed, and data generated are clear.

A number of issues raised earlier were addressed by the authors in the revised manuscript. The responses are convincing. These included circadian rhythm considerations, baroreflex findings, BP fluctuations driven by animal movement, and data presentation.

Overall, this is a solid study with huge physiological implications. I believe that it will be of great benefit to the field.

---

## [Author Response]

The following is the authors’ response to the original reviews

**Public Reviews:**

**Reviewer #1 (Public review):**
This study examined the effect of blood pressure variability on brain microvascular function and cognitive performance. By implementing a model of blood pressure variability using an intermittent infusion of AngII for 25 days, the authors examined different cardiovascular variables, cerebral blood flow, and cognitive function during midlife (12-15-month-old mice). Key findings from this study demonstrate that blood pressure variability impairs baroreceptor reflex and impairs myogenic tone in brain arterioles, particularly at higher blood pressure. They also provide evidence that blood pressure variability blunts functional hyperemia and impairs cognitive function and activity. Simultaneous monitoring of cardiovascular parameters, in vivo imaging recordings, and the combination of physiological and behavioral studies reflect rigor in addressing the hypothesis. The experiments are well-designed, and the data generated are clear. I list below a number of suggestions to enhance this important work:(1) Figure 1B: It is surprising that the BP circadian rhythm is not distinguishable in either group. Figure 2, however, shows differences in circadian rhythm at different timepoints during infusion. Could the authors explain the lack of circadian effect in the 24-h traces?

The circadian rhythm pattern is apparent in Figure 2 (Active BP higher than Inactive BP), where BP is presented as 12hour averages. When the BP data is expressed as one-hour averages (rather than minute-to-minute) over 24hours, now included in the revised manuscript as Supplemental Figure 3C-D, the circadian rhythm becomes noticeable. In addition, we have included one-hour average BP data for all mice in the control and BPV groups, Supplemental Figure 3A-B.

Notably, the Ang-II induced pulsatile BP pattern remains evident in the one-hour averages for the BPV group, Supplemental Figure 3B. To minimize bias and validate variability, pump administrations start times were randomized for both control and BPV groups, Supplemental Figure 3A-B. Despite these adjustments, the circadian rhythm profile of BP is consistently maintained across individual mice and in the collective dataset, Supplemental Figure 3C-D.

(2) While saline infusion does not result in elevation of BP when compared to Ang II, there is an evident "and huge" BP variability in the saline group, at least 40mmHg within 1 hour. This is a significant physiological effect to take into consideration, and therefore it warrants discussion.

Thank you for this comment. The large variations in BP in the raw traces during saline infusion reflects transient BP changes induced by movement/activity, which is now included in Figure 1B (maroon trace). The revised manuscript now includes Line 222 “Note that dynamic activity-driven BP changes were apparent during both saline- and Ang II infusions, Figure 1B”.

(3) The decrease in DBP in the BPV group is very interesting. It is known that chronic Ang II increases cardiac hypertrophy, are there any changes to heart morphology, mass, and/or function during BPV? Can the decrease in DBP in BPV be attributed to preload dysfunction? This observation should be discussed.

The lower DBP in the BPV group was already present at baseline, while both groups were still infused with saline, and was a difference beyond our control. However, this is an important and valid consideration, particularly considering the minimal yet significant increase in SBP within the BPV group (Figure 1D). Our goal was to induce significant transient blood pressure responses (BPV) and investigate the impact on cardiovascular and neurovascular outcomes in the absence of hypertension. We did not anticipate any major cardiac remodeling at this early time point (considering the absence of overt hypertension) and thus cardiac remodeling was not assessed and this is now discussed in the revised manuscript (Line 443-453).

(4) Examining the baroreceptor reflex during the early and late phases of BPV is quite compelling. Figures 3D and 3E clearly delineate the differences between the two phases. For clarity, I would recommend plotting the data as is shown in panels D and E, rather than showing the mathematical ratio. Alternatively, plotting the correlation of ∆HR to ∆SBP and analyzing the slopes might be more digestible to the reader. The impairment in baroreceptor reflex in the BPV during high BP is clear, is there any indication whether this response might be due to loss of sympathetic or gain of parasympathetic response based on the model used?

We appreciate the reviewer’s suggestion and have accordingly generated new figures displaying scatter plots of SBP vs HR with linear regression analysis (Figure 3D-G). Our goal is to further investigate which branch of the autonomic nervous system is affected in this model. The loss of a bradycardic response suggests either an enhancement of sympathetic activity, a reduction in parasympathetic activity, or a combination of both. This is briefly discussed in the revised manuscript (Line 486-496).

Heart rate variability (HRV) serves as an index of neurocardiac function and dynamic, non-linear autonomic nervous system processes, as described in Shaffer and Ginsber[1]. However, given that our data was limited to BP and HR readings collected at one-minute intervals, our primary assessment of autonomic function is limited to the bradycardic response. Further studies will be necessary to fully characterize the autonomic parameters influenced by chronic BPV.

(5) Figure 3B shows a drop in HR when the pump is ON irrespective of treatment (i.e., independent of BP changes). What is the underlying mechanism?

We apologize for any lack of clarity. These observed heart rate (HR) changes occurred during Ang II infusion, when blood pressure (BP) was actively increasing. In the control group, the pump solution was switched to Ang II during specific periods (days 3-5 and 21-25 of the treatment protocol) to induce BP elevations and a baroreceptor response, allowing direct comparisons between the control and BPV group.

To clarify this point, we have revised Line 260-263 of the manuscript: “To compare pressure-induced bradycardic responses between BPV and control mice at both early and later treatment stages, a cohort of control mice received Ang II infusion on days 3-5 (early phase) (Supplemental Figure 4) and days 21-25 (late phase) thereby transiently increasing BP”.

Additionally, a detailed description has been added to the Methods section (Line 96-101): “Controls receiving Ang II: To facilitate between-group comparisons (control vs BPV), a separate cohort of control mice were subjected to the same pump infusion parameters as BPV mice but for a brief period receiving Ang II infusions on days 3-5 and 21-25 for experiments assessing pressure-evoked responses, including bradycardic reflex, myogenic response, and functional hyperemia at high BP.”

(6) The correlation of ∆diameter vs MAP during low and high BP is compelling, and the shift in the cerebral autoregulation curve is also a good observation. I would strongly recommend that the authors include a schematic showing the working hypothesis that depicts the shift of the curve during BPV.

Thank you for this insightful comment. The increase in vessel reactivity to BP elevations in parenchymal arterioles of BPV mice suggests that chronic BPV induces a leftward shift and a potential narrowing of the cerebral autoregulation range (lower BP thresholds for both the upper and lower limits of autoregulation). This has been incorporated (and discussed) into the revised manuscript (see Figure 5N).

One potential explanation for these changes is that the absence of sustained hypertension, a prominent feature in most rodent models of hypertension, limits adaptive processes that protect the cerebral microcirculation from large BP fluctuations (e.g., vascular remodeling). While this study does not specifically address arteriole remodeling, the lack of such adaptation may reduce pressure buffering by upstream arterioles, thereby rendering the microcirculation more vulnerable to significant BP fluctuations.

The unique model allows for measurements of parenchymal arteriole reactivity to acute dynamic changes in BP (both an increase and decrease in MAP). Our findings indicate that chronic BPV enhances the reactivity of parenchymal arterioles to BP changes—both during an increase in BP and upon its return to baseline, Supplemental Figure 5C, F. The data suggest an increased myogenic response to pressure elevation, indicative of heightened contractility, a common adaptive process observed in rodent models of hypertension[2-4]. However, our model also reveals a notable tendency for greater dilation when the BP drops, Supplemental Figure 5F. This intriguing observation may suggest ischemia during the vasoconstriction phase (at higher BP), leading to enhanced release of dilatory signals, which subsequently manifest as a greater dilation upon BP reduction. This phenomenon bears similarities to chronic hypoperfusion models[5,6], where vasodilatory mechanisms become more pronounced in response to sustained ischemic conditions. Future studies investigating the effects of BPV on myogenic responses and brain perfusion will be a priority for our ongoing research.

(7) Functional hyperemia impairment in the BPV group is clear and well-described. Pairing this response with the kinetics of the recovery phase is an interesting observation. I suggest elaborating on why BPV group exerts lower responses and how this links to the rapid decline during recovery.

Based on the heightened reactivity of BPV parenchymal arterioles to intravascular pressure (Figure 5), we anticipate that the reduction of sensory-evoked dilations results from an increased vasoconstrictive activity and/or a decreased availability of vasodilatory signaling pathways (NO, EETs, COX-derived prostaglandins)[7,8]. Consequently, the magnitude of the FH response is blunted during periods of elevated BP in BPV mice.

Additionally, upon termination of the stimulus-induced response−when vasodilatory signals would typically dominate−vasoconstrictive mechanisms are rapidly engaged (or unmasked), leading to quicker return to baseline. This shift in the balance between vasodilatory and vasoconstrictive forces favors vasoconstriction, contributing to the altered recovery kinetics observed in BPV mice. This has been included in the Discussion section of the revised manuscript.

(8) The experimental design for the cognitive/behavioral assessment is clear and it is a reasonable experiment based on previous results. However, the discussion associated with these results falls short. I recommend that the authors describe the rationale to assess recognition memory, short-term spatial memory, and mice activity, and explain why these outcomes are relevant in the BPV context. Are there other studies that support these findings? The authors discussed that no changes in alternation might be due to the age of the mice, which could already exhibit cognitive deficits. In this line of thought, what is the primary contributor to behavioral impairment? I think that this sentence weakens the conclusion on BPV impairing cognitive function and might even imply that age per se might be the factor that modulates the various physiological outcomes observed here. I recommend clarifying this section in the discussion.

We thank the reviewer for this comment. Clinical studies have demonstrated that patients with elevated BPV exhibit impairments across multiple cognitive domains, including declines in processing speed[9] and episodic memory[10]. To evaluate memory function, we utilized behavioral tests: the novel object recognition (NOR) task to assess episodic memory[11] and the spontaneous Y-maze to evaluate short-term spatial memory[12].

Previous research indicates that older C57Bl6 mice (14-month-old) exhibit cognitive deficits compared to younger counterparts (4- and 9-month-old)[13]. To ensure rigorous selection for behavioral testing, we conducted preliminary *NOR* assessment, evaluating recognition memory at the one-hour delay but observing failures at the four-, and 24-hour delays, indicating age-related deficits. Based on these results, animals failing recognition criteria were excluded from subsequent behavioral assessment. However, because no baseline cognitive testing was conducted for the spontaneous Y-maze, it is possible that some mice with aged-related deficits were included in this test, which may have influenced data interpretation.

Additionally, the absence of differences in the Y-maze performance may suggest that short-term spatial memory remains intact following 25 days of BPV, a point that is now discussed in the revised manuscript.

(9) Why were only male mice used?

We appreciate this comment and acknowledge the importance of conducting experiments in both male and female mice. Studies involving female mice are currently ongoing, with telemetry data collection approximately halfway completed and two-photon imaging studies on functional hyperemia also partially completed. However, using middleaged mice for these experiments has proven challenging due to high mortality rates following telemetry surgeries. As a result, we initially limited our first cohort to male mice.

(10) In the results for Figure 3: "Ang II evoked significant increases in SBP in both control and BPV groups;...". Also, in the figure legend: "B. Five-minute average HR when the pump is OFF or ON (infusing Ang II) for control and BPV groups...." The authors should clarify this as the methods do not state a control group that receives Ang II.

Please refer to response to comment 5.

**Reviewer #2 (Public review):**
Summary:Blood pressure variability has been identified as an important risk factor for dementia. However, there are no established animal models to study the molecular mechanisms of increased blood pressure variability. In this manuscript, the authors present a novel mouse model of elevated BPV produced by pulsatile infusions of high-dose angiotensin II (3.1ug/hour) in middle-aged male mice. Using elegant methodology, including direct blood pressure measurement by telemetry, programmable infusion pumps, in vivo two-photon microscopy, and neurobehavioral tests, the authors show that this BPV model resulted in a blunted bradycardic response and cognitive deficits, enhanced myogenic response in parenchymal arterioles, and a loss of the pressure-evoked increase in functional hyperemia to whisker stimulation.Strengths:As the presentation of the first model of increased blood pressure variability, this manuscript establishes a method for assessing molecular mechanisms. The state-of-the-art methodology and robust data analysis provide convincing evidence that increased blood pressure variability impacts brain health.Weaknesses:One major drawback is that there is no comparison with another pressor agent (such as phenylephrine); therefore, it is not possible to conclude whether the observed effects are a result of increased blood pressure variability or caused by direct actions of Ang II.

We acknowledge this limitation and have attempted to address the concern by introducing an alternative vasopressor, norepinephrine (NE), Figure 4. A subcutaneous dose of 45 µg/kg/min was titrated to match Ang II-induced transient BP pulse (Systolic BP ~150-180 mmHg), Figure 4A. Similar to Ang II treated mice, NE-treated mice exhibited no significant changes in average mean arterial pressure (MAP) throughout the 20-day treatment period (Figure 4B). Although there was a trend (P=0.08) towards increased average real variability (ARV) (Figure 4C left), it did not reach statistical significance. The coefficient of variation (CV) (Figure 4C right) was significantly increased by day 3-4 of treatment (P=0.02).

Notably, unlike the bradycardic response observed during Ang II-induced BP elevations, NE infusions elicited a tachycardic response (Figure 4A), likely due to β-1 adrenergic receptor activation. However, significant mortality was observed within the NE cohort: three of six mice died prematurely during the second week of treatment, and two additional mice required euthanasia on days 18 and 20 due to lethargy, impaired mobility, and tachypnea.

While we recognize the importance of comparing results across vasopressors, further investigation using additional vasopressors would require a dedicated study, as each agent may induce distinct off-target effects, potentially generating unique animal models. Alternatively, a mechanical approach−such as implanting a tethered intra-aortic balloon[14] connected to a syringe pump−could be explored to modulate blood pressure variability without pharmacological intervention. However, such an approach falls beyond the scope of the present study.

Ang II is known to have direct actions on cerebrovascular reactivity, neuronal function, and learning and memory. Given that Ang II is increased in only 15% of human hypertensive patients (and an even lower percentage of non-hypertensive), the clinical relevance is diminished. Nonetheless, this is an important study establishing the first mouse model of increased BPV.

We agree that high Ang II levels are not a predominant cause of hypertension in humans, which is why it is critical that our pulsatile Ang II dosing did not cause overt hypertension, (no increase in 24-hour MAP). Ang II was solely a tool to produce controlled, transient increases in BP to yield a significant increase in BPV.

Regarding BPV specifically, prior studies indicate that primary hypertensive patients with elevated urinary angiotensinogen-to-creatinine ratio exhibit significantly higher mean 24-hour systolic ARV compared to those with lower ratios[15]. However, the fundamental mechanisms driving these harmful increases in BPV remain poorly defined. A central theme across clinical BPV studies is impaired arterial stiffness, which has been proposed to contribute to BPV through reduced arterial compliance and diminished baroreflex sensitivity. Moreover, increased BPV can exert mechanical stress on arterial walls, leading to arterial remodeling and stiffness−ultimately perpetuating a detrimental feed-forward cycle[16].

In our model, male BPV mice exhibited a minimal yet significant elevation in SBP without corresponding increases in DBP, potentially reflecting isolated systolic hypertension, which is strongly associated with arterial stiffness[17,18]. Our initial goal was to establish controlled rapid fluctuations in BP, and Ang II was selected as the pressor due to its potent vasoconstrictive properties and short half-life[19].

We appreciate the reviewer’s insightful comment and acknowledge the necessity of exploring alternative mechanisms underlying BPV, and independent of Ang II. It is our long-term goal to investigate these factors in further studies.

**Recommendations for the authors:**

**Reviewer #2 (Recommendations for the authors):**
(1) How was the dose of Ang II determined? It seems that this dose (3.1ug/hr) is quite high.

The Ang II dose was titrated in a preliminary study to one that induced a significant and transient BP response without increasing 24-hour blood pressure (i.e. no hypertension).

Ang II was delivered subcutaneously at 3.1 μg/hr, a concentration comparable to high-dose Ang II administration via mini-osmotic pumps (~1700 ng/kg/min)[20], with one-hour pulses occurring every 3-4 hours. With 6 pulses per day, the total daily dose equates to 18.6 µg/day in a ~30 gram mouse.

For comparison, if the same 18.6 µg/day dose were administered continuously via a mini-osmotic pump (18.6 µg/0.03kg/1440min), the resulting dosage would be approximately 431 ng/kg/min[21,22], aligning with subpressor dose levels. Thus, while the total dose may appear high, it is not delivered in a constant manner but rather intermittently, allowing for controlled, rapid variations in blood pressure.

(2) Were behavioral studies performed on the same mice that were individually housed? Individual housing causes significant stress in mice that can affect learning and memory tasks (PMC6709207). It's not a huge issue since the control mice would have been housed the same way, but it is something that could be mentioned in the discussion section.

Behavioral studies were performed on mice that were individually housed following the telemetry surgery. The study was started once BP levels stabilized, as mice required several days to achieve hemodynamic stability post-surgery. Consequently, all mice were individually housed for several days before undergoing behavioral assessment.

To account for potential cognitive variability, earlier novel object recognition (NOR) tests were conducted to established cognitive capacity, and mice that did not meet criteria were excluded from further behavioral testing. However, we acknowledge that individual housing induces stress, which can influence learning and memory, and this is a factor we were unable to fully control. Given that both experimental and control groups experienced the same housing conditions, this stress effect should be comparable across cohorts. A discussion on this limitation is now included in the text.

(3) It looks like one control mouse that was included in both Figures 1 and 2 (control n=12) but was excluded in Table 1 (control n=11), this isn't mentioned in the text - please include the exclusion criteria in the manuscript.

We apologize for the typo−12 control animals were consistently utilized across Figure 1-2, Table 1, Supplemental Table 1, Figure 6C, and Supplemental Figure 2B. Since the initial submission, one control mouse was completed and included into the telemetry control cohort. Thus, in the updated manuscript, we have corrected the control sample size to 13 mice across these figures ensuring consistency.

Additionally, exclusion criteria have now been explicitly included in the manuscript (Line 173-175). Mice were excluded from the study if they died prematurely (died prior to treatment onset) or mice exhibited abnormally elevated pressure while receiving saline, likely due to complications from telemetry surgery.

(4) Please include a statement on why female mice were not included in this study.

As discussed in our response to Reviewer #1, our initial intention was to include both male and female mice in this study. However, high mortality rates following telemetry surgeries significantly constrained our ability to advance all aspects of the study. As a result, we limited our first cohort to males to establish the basics of the model. A statement is now included in the manuscript, Line 50-53: “Female mice were not included in the present study due to high post-surgery mortality observed in 12-14-month-old mice following complex procedures. To minimized confounding effects of differential survival and to establish foundational data for this model, we restricted the investigation to male mice.”

Potential sex differences might be complex and warrants a separate future research to comprehensively assess sex as a biological variable, which are currently ongoing.

(5) On page 14, "experiments from control vs experimental mice were not equally conducted in the same season raising the possibility for a seasonal effect" - does this mean that control experiments were not conducted at the same time as the Ang II infusions in BPV mice? This has huge implications on whether the effects observed are induced by treatment or just batch seasonal effects.

We fully acknowledge the reviewer’s concern, and our statement aims to provide transparency regarding the study’s limitations. Several challenges contributed to this outcome, including high mortality rates following surgeries (primarily telemetry implantation) and technical issues related to instrumentation, particularly telemetry functionality.

Differences between BPV and saline mice emerge primarily due to mortality or telemetry failures−some mice did not survive post-surgery, while others remain healthy but had non-functional telemeters. This issue was particularly pronounced in 14-month-old mice, as their fragile vasculature occasionally prevented proper BP readings.

Each experiment required a minimum of two and a half months per mouse to complete, with a cost (also per mouse) exceeding $1500 USD ($300 pump, $175 mouse, $900 telemeters, per diem, drugs, reagents etc.). Despite our best effort to ensure comparable seasonal/batch data, these logistical and technical constraints prevented perfect synchronization.

To evaluate whether seasonal differences influenced our results, we incorporated additional telemetry data into the control cohort. Of the seven included control mice, six underwent the same treatment but were allocated to a separate branch of the study, which endpoints did not require a chronic cranial window. We found no significant differences in 24-hour average MAP during the baseline period between control mice with or without a cranial window, Supplemental Figure 2A. Additionally, we grouped mice into seasonal categories based on Georgia’s climate: “Spring-Summer” (May-September) and “Fall-Winter” (October-April) but observed no BP differences between these periods, Supplemental Figure 2B.

Given the absence of seasonal effects on BP and the fact that mice were sourced from two independent suppliers (Jackson Laboratory and NIA), we anticipate that the observed results are driven by treatment rather than seasonal or batch effects.

(6) Methods, two-photon imaging: did the authors mean "retro-orbital" instead of "intra-orbital" injection of the Texas red dye? Also, is this a Texas red-dextran? If so, what molecular weight?

Thank you for this comment. The correct terminology is “retro-orbital” rather than “intra-orbital” injection. Additionally, we utilized Texas Red-dextran (70 kDa, 5% [wt/vol] in saline) for the imaging experiments. These details have now been incorporated into the Methods section.

(1) Shaffer F, Ginsberg JP. An Overview of Heart Rate Variability Metrics and Norms. Front Public Health. 2017;5:258. doi: 10.3389/fpubh.2017.00258

(2) Pires PW, Jackson WF, Dorrance AM. Regulation of myogenic tone and structure of parenchymal arterioles by hypertension and the mineralocorticoid receptor. Am J Physiol Heart Circ Physiol. 2015;309:H127-136. doi: 10.1152/ajpheart.00168.2015

(3) Iddings JA, Kim KJ, Zhou Y, Higashimori H, Filosa JA. Enhanced parenchymal arteriole tone and astrocyte signaling protect neurovascular coupling mediated parenchymal arteriole vasodilation in the spontaneously hypertensive rat. J Cereb Blood Flow Metab. 2015;35:1127-1136. doi: 10.1038/jcbfm.2015.31

(4) Diaz JR, Kim KJ, Brands MW, Filosa JA. Augmented astrocyte microdomain Ca(2+) dynamics and parenchymal arteriole tone in angiotensin II-infused hypertensive mice. Glia. 2019;67:551-565. doi: 10.1002/glia.23564

(5) Kim KJ, Diaz JR, Presa JL, Muller PR, Brands MW, Khan MB, Hess DC, Althammer F, Stern JE, Filosa JA. Decreased parenchymal arteriolar tone uncouples vessel-to-neuronal communication in a mouse model of vascular cognitive impairment. GeroScience. 2021. doi: 10.1007/s11357-020-00305-x

(6) Chan SL, Nelson MT, Cipolla MJ. Transient receptor potential vanilloid-4 channels are involved in diminished myogenic tone in brain parenchymal arterioles in response to chronic hypoperfusion in mice. Acta Physiol (Oxf). 2019;225:e13181. doi: 10.1111/apha.13181

(7) Tarantini S, Hertelendy P, Tucsek Z, Valcarcel-Ares MN, Smith N, Menyhart A, Farkas E, Hodges EL, Towner R, Deak F, et al. Pharmacologically-induced neurovascular uncoupling is associated with cognitive impairment in mice. J Cereb Blood Flow Metab. 2015;35:1871-1881. doi: 10.1038/jcbfm.2015.162

(8) Ma J, Ayata C, Huang PL, Fishman MC, Moskowitz MA. Regional cerebral blood flow response to vibrissal stimulation in mice lacking type I NOS gene expression. Am J Physiol. 1996;270:H1085-1090. doi: 10.1152/ajpheart.1996.270.3.H1085

(9) Sible IJ, Nation DA. Blood Pressure Variability and Cognitive Decline: A Post Hoc Analysis of the SPRINT MIND Trial. Am J Hypertens. 2023;36:168-175. doi: 10.1093/ajh/hpac128

(10) Epstein NU, Lane KA, Farlow MR, Risacher SL, Saykin AJ, Gao S. Cognitive dysfunction and greater visit-to-visit systolic blood pressure variability. Journal of the American Geriatrics Society. 2013;61:2168-2173. doi: 10.1111/jgs.12542

(11) Antunes M, Biala G. The novel object recognition memory: neurobiology, test procedure, and its modifications. Cognitive processing. 2012;13:93-110. doi: 10.1007/s10339-011-0430-z

(12) Kraeuter AK, Guest PC, Sarnyai Z. The Y-Maze for Assessment of Spatial Working and Reference Memory in Mice. Methods Mol Biol. 2019;1916:105-111. doi: 10.1007/978-1-4939-8994-2_10

(13) Singhal G, Morgan J, Jawahar MC, Corrigan F, Jaehne EJ, Toben C, Breen J, Pederson SM, Manavis J, Hannan AJ, et al. Effects of aging on the motor, cognitive and affective behaviors, neuroimmune responses and hippocampal gene expression. Behav Brain Res. 2020;383:112501. doi: 10.1016/j.bbr.2020.112501

(14) Tediashvili G, Wang D, Reichenspurner H, Deuse T, Schrepfer S. Balloon-based Injury to Induce Myointimal Hyperplasia in the Mouse Abdominal Aorta. J Vis Exp. 2018. doi: 10.3791/56477

(15) Ozkayar N, Dede F, Akyel F, Yildirim T, Ates I, Turhan T, Altun B. Relationship between blood pressure variability and renal activity of the renin-angiotensin system. J Hum Hypertens. 2016;30:297-302. doi: 10.1038/jhh.2015.71

(16) Kajikawa M, Higashi Y. Blood pressure variability and arterial stiffness: the chicken or the egg? Hypertens Res. 2024;47:1223-1224. doi: 10.1038/s41440-024-01589-8

(17) Laurent S, Boutouyrie P. Arterial Stiffness and Hypertension in the Elderly. Front Cardiovasc Med. 2020;7:544302. doi: 10.3389/fcvm.2020.544302

(18) Wallace SM, Yasmin, McEniery CM, Maki-Petaja KM, Booth AD, Cockcroft JR, Wilkinson IB. Isolated systolic hypertension is characterized by increased aortic stiffness and endothelial dysfunction. Hypertension. 2007;50:228-233. doi: 10.1161/HYPERTENSIONAHA.107.089391

(19) Al-Merani SA, Brooks DP, Chapman BJ, Munday KA. The half-lives of angiotensin II, angiotensin II-amide, angiotensin III, Sar1-Ala8-angiotensin II and renin in the circulatory system of the rat. J Physiol. 1978;278:471490. doi: 10.1113/jphysiol.1978.sp012318

(20) Zimmerman MC, Lazartigues E, Sharma RV, Davisson RL. Hypertension caused by angiotensin II infusion involves increased superoxide production in the central nervous system. Circ Res. 2004;95:210-216. doi: 10.1161/01.RES.0000135483.12297.e4

(21) Gonzalez-Villalobos RA, Seth DM, Satou R, Horton H, Ohashi N, Miyata K, Katsurada A, Tran DV, Kobori H, Navar LG. Intrarenal angiotensin II and angiotensinogen augmentation in chronic angiotensin II-infused mice. Am J Physiol Renal Physiol. 2008;295:F772-779. doi: 10.1152/ajprenal.00019.2008

(22) Nakagawa P, Nair AR, Agbor LN, Gomez J, Wu J, Zhang SY, Lu KT, Morgan DA, Rahmouni K, Grobe JL, et al. Increased Susceptibility of Mice Lacking Renin-b to Angiotensin II-Induced Organ Damage. Hypertension. 2020;76:468-477. doi: 10.1161/HYPERTENSIONAHA.120.14972